# CYRI controls epidermal wound closure and cohesion of invasive border cell cluster in *Drosophila*

Marvin Rötte[1]*, Mila Y. Höhne[1]*, Dennis Klug[1], Kirsten Ramlow[1], Caroline Zedler[1], Franziska Lehne[1], Meike Schneider[1], Maik C. Bischoff[1], and Sven Bogdan[1]

**Cell motility is crucial for many biological processes including morphogenesis, wound healing, and cancer invasion. The WAVE regulatory complex (WRC) is a central Arp2/3 regulator driving cell motility downstream of activation by Rac GTPase. CYFIP-related Rac1 interactor (CYRI) proteins are thought to compete with WRC for interaction with Rac1 in a feedback loop regulating lamellipodia dynamics. However, the physiological role of CYRI proteins in vivo in healthy tissues is unclear. Here, we used *Drosophila* as a model system to study CYRI function at the cellular and organismal levels. We found that CYRI is not only a potent WRC regulator in single macrophages that controls lamellipodial spreading but also identified CYRI as a molecular brake on the Rac-WRC-Arp2/3 pathway to slow down epidermal wound healing. In addition, we found that CYRI limits invasive border cell migration by controlling cluster cohesion and migration. Thus, our data highlight CYRI as an important regulator of cellular and epithelial tissue dynamics conserved across species.**

## Introduction

Cell migration has a key role during tissue morphogenesis, tissue repair, wound healing, and cancer spreading (Friedl and Gilmour, 2009; Merino et al., 2020; Schaks et al., 2019; Svitkina, 2018). Cells can either migrate individually or in cohesive groups depending on the cell type and the three-dimensional environment (Yamada and Sixt, 2019). Both, single-cell migration and collective-cell migration require actin polymerization providing the mechanical forces to drive membrane protrusions (Pollard, 2022; SenGupta et al., 2021). The most prominent actin structures mediating the cell protrusion are lamellipodia and filopodia, which are stimulated by Rho-family GTPases, especially Rac1 and Cdc42 (Ridley, 2015). Lamellipodia are an archetypal type of flat membrane protrusions found at the leading edge of diverse cells including crawling immune cells, epithelial cell sheets, and invasive cell clusters (Jacinto et al., 2001; Machesky, 2008; Small et al., 2002). The formation of lamellipodial protrusion is driven by the assembly of branched actin networks, initiated by the branched actin nucleator Arp2/3 complex (Goley and Welch, 2006; Krause and Gautreau, 2014; Pollard and Borisy, 2003). The Arp2/3 complex itself must be activated by the WAVE regulatory complex (WRC) (Bieling and Rottner, 2023; Chen et al., 2010; Schaks et al., 2019). The WRC consists of five subunits: CYFIP/Sra-1, NAP1/Kette, WAVE, Abi, and HSPC300. WAVE is transinhibited by this complex, and this inhibition can be released by Rac binding (Chen et al., 2010). The Rac binding surfaces on the WRC are formed by the CYFIP/Sra-1 subunit which sequesters the C-terminal Arp2/3 activating the WCA domain of WAVE from accessing the Arp2/3 complex (Chen et al., 2010; Ding et al., 2022; Oikawa et al., 2004).

Rac1, WRC, and Arp2/3 form a feedback loop to allow dynamic actin filament reorganization, controlled by many other signaling and actin-binding proteins (Chen et al., 2014a, 2014b; Yang et al., 2022). Such feedback control also includes regulatory proteins such as gadkin and arpin, which directly bind and inhibit Arp2/3 function (Chánez-Paredes et al., 2019; Gautreau et al., 2022). More recently, a new class of conserved Rac effectors related to CYFIP/Sra-1 has been identified as negative regulators of the feedback loop for branched network assembly (Chattaragada et al., 2018; Fort et al., 2018; Shang et al., 2018; Yuki et al., 2019). These CYFIP-related RAC1 interacting proteins (CYRIs) are thought to compete with the WRC for active Rac1, thereby locally suppressing WRC-Arp2/3-dependent branched actin nucleation in lamellipodial protrusions (Fort et al., 2018; Whitelaw et al., 2019; Yuki et al., 2019). In mammals, there are two differentially expressed isoforms with high sequence identity, CYRI-A and CYRI-B, which might have distinct developmental and physiological functions in vivo (Machesky, 2023). Biochemically, both isoforms interact specifically with active Rac1, whereby CYRI-A has a higher affinity than CYRI-B (Le et al., 2021). At the cellular level, CYRI-A and CYRI-B have clear overlapping functions and can compensate for each other

---

[1]Department of Molecular Cell Physiology, Institute of Physiology and Pathophysiology, Philipps-University Marburg, Marburg, Germany.

*M. Rötte and M.Y. Höhne contributed equally to this paper. Correspondence to Sven Bogdan: sven.bogdan@staff.uni-marburg.de.

in WRC-dependent cell motility and macropinocytosis (Le et al., 2021).

While the function of CYRI proteins in single cells has been studied in more detail, the function of CYRI proteins in healthy tissues and model organisms has not been addressed yet (Machesky, 2023; Whitelaw et al., 2019). Here, we analyzed the single *Drosophila* gene CG32066 encoding the homolog of mammalian CYRI-A/CYRI-B. Like its mammalian counterparts, *Drosophila* CYRI preferentially binds activated Rac1 and suppresses lamellipodial protrusions upon overexpression. Flies lacking CYRI function are viable but partially sterile. Loss- and gain-of-function analysis further revealed important functions in lamellipodial protrusions during epidermal wound closure and in invasive, collective border cell migration. Thus, our data highlight the evolutionary conserved role of CYRI proteins in regulating cellular and tissue dynamics.

## Results

### CG32066 encodes the *Drosophila* homolog of CYRI, a protein interacting with activated Rac1

Sequence analysis revealed that the gene *CG32066* encodes a member of the CYRI protein family, a new group of Rac1 interactors recently described (Whitelaw et al., 2019). The AlphaFold algorithm (Jumper et al., 2021) predicts a highly similar structure and topology between CG32066 and human CYRI-A and CYRI-B proteins (Fig. 1, A–A″ and Video 1). *Drosophila* CYRI also contains a domain of unknown function (DUF1394, Fig. S1 A) that has been recently characterized as a Rac-binding module mediated by two highly conserved arginines at positions 163 and 164 (homologous to arginine R160 and R161 in CYRI-A and CYRI-B; Whitelaw et al., 2019; Fig. 1, A and B highlighted in yellow). The *Drosophila* CG32066 protein exhibits ∼55% identity with its human homolog CYRI-B (Fig. 1 C and Fig. S1 A′).

We first tested whether *Drosophila* CYRI could indeed interact with active Rac1. We performed pull-down experiments using GST-tagged proteins, either wildtype GST-CYRI$^{WT}$ or mutant GST-CYRI$^{R163/164D}$ with mutations of key arginines (R163D, R164D; see also Fig. S1 A), incubated with cell lysates expressing HA-tagged dominant-negative Rac1$^{N17}$ or constitutively active Rac1$^{V12}$ proteins (Fig. 1 D). Recombinant wildtype CYRI protein significantly bound the active Rac1$^{V12}$ variant, whereas mutation of key arginines to aspartic acid in CYRI abrogated this interaction (Fig. 1 D; quantification in Fig. 1 E). We also performed similar pull-down experiments using GST-tagged wildtype Rac1 (Rac-WT) and constitutively active Rac, RacQ61L, incubated either with cell lysates expressing HA-tagged wildtype CYRI (HA-CYRI-WT) or the mutant HA-tagged CYRI$^{R163/164D}$ (Fig. S1 B). CYRI-WT interacted more strongly with RacQ61L compared with Rac-WT as previously shown for mammalian CYRI-B (Fort et al., 2018). Again, mutations of key arginines to aspartic acid in CYRI abrogated this interaction (Fig. S1 B).

We also used the bimolecular fluorescence complementation (BiFC) assay to further evaluate a direct interaction between CYRI and Rac1 in vivo (Gohl et al., 2010). Upon coexpression of an N-terminal Rac1-NYFP and C-terminal CYRI-CYFP fusion under the control of the *en*-Gal4 driver, strong YFP fluorescence

was observed in the posterior compartment of wing discs (Fig. 1 F). Mutation of key arginines in CYRI strongly reduced the BiFC signal in wing imaginal discs (Fig. 1 G). Taken together, these data show that *Drosophila* CYRI is a conserved Rac1 interacting protein.

### *Drosophila* CYRI inhibits lamellipodial protrusions

An affinity-purified antibody to endogenous CYRI did not give specific staining by immunofluorescence (Fig. S1, C and D) but detected endogenous protein of about 35 kDa in western blots of lysates from *Drosophila* S2 cells (Fig. 2 A). To visualize CYRI in living cells, we employed CRISPR-mediated homology-directed repair to insert the GFP coding sequence at the 3′ end of the *cyri* gene to visualize CYRI localization in living S2 cells. Western blot analysis from this clone-selected stable S2 cell line (termed CYRI-GFP *k-in*) detected the endogenous CYR-GFP fusion protein at the expected size of 70 kDa, both using an anti-CYRI and an anti-GFP antibody (Fig. 2 A). Adding a GFP tag to endogenous CYRI did not affect S2 cell morphology or cell size as previously reported in cultured mammalian cell culture ectopically expressing GFP-CYRI fusions (Fig. 2 B; [Fort et al., 2018]). Despite the very low endogenous GFP fluorescence, confocal fluorescence microscopy revealed a distinct enrichment of the endogenous CYR-GFP protein at leading pseudopods, both in fixed and in living cultured S2 cells (Fig. 2, C and D; and Video 2). Time-lapse fluorescence microscopy further revealed a localization of endogenous CYRI to macropinocytic cups and macropinocytic structures as previously found for mammalian CYRI-A protein (Fig. 2 E and Video 2; [Le et al., 2021]).

Overexpression of CYRI proteins suppresses lamellipodia protrusions phenocopying wave mutant cell morphology (Fort et al., 2018; Whitelaw et al., 2019; Yuki et al., 2019). To determine the functional activity of *Drosophila* CYRI on protruding lamellipodia, we performed transient overexpression experiments using cultured *Drosophila* S2R+ cells. Non-transfected cells or control S2R+ cells expressing GFP showed broad circumferential lamellipodia when spread on Concanavalin A (ConA) (Fig. 2 F; Bogdan et al., 2005). By contrast, overexpression of wildtype CYRI led to a collapse of all lamellipodia-like structures and induced a spiky morphology (Fig. 2 G). Quantitative analysis confirmed that CYRI overexpression phenocopied a loss of wave function (Fig. 2 I; Bogdan et al., 2005). For overexpression of wildtype CYRI (CYRI-WT), about 80% of all transfected cells identified by anti-CYRI staining showed a spiky phenotype. Mutation of key arginines in CYRI$^{R163/164D}$ abolished this phenotype upon overexpression. The majority of cells expressing CYRI$^{R163/164D}$ showed a wildtype spread cell morphology and only 15% showed changed morphology similar to control cells transfected with a GFP control construct (Fig. 2 H, quantification in Fig. 2 I). Thus, we conclude that Rac1-binding is needed to competitively inhibit WAVE function in lamellipodia formation.

### Loss of CYRI function promotes lamellipodial spreading through increased WAVE localization

We next analyzed the consequences of loss of *cyri* function. The *Drosophila cyri* gene is located on the third chromosome (Fig. 3 A). It consists of four exons encoding a 36.6 kDa protein. We took

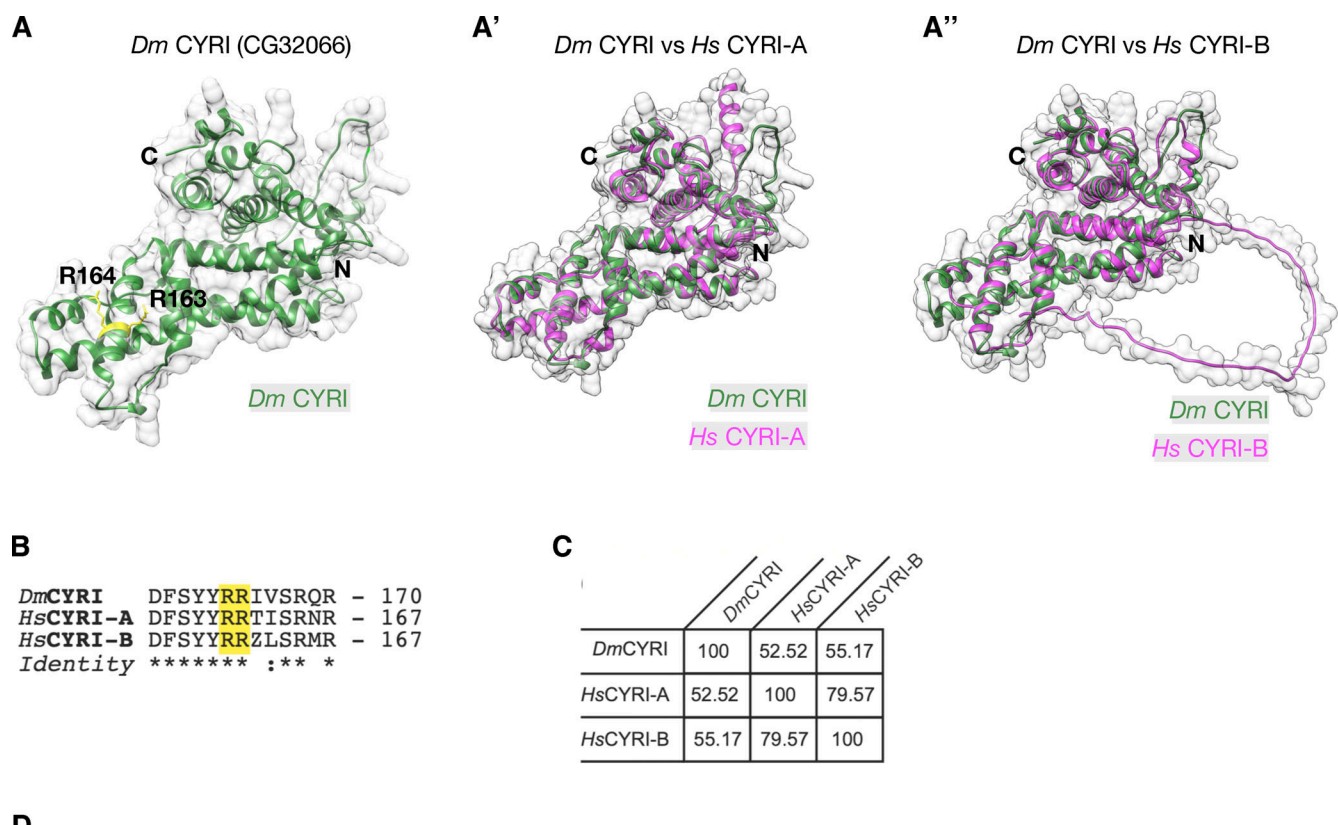

**A** *Dm* CYRI (CG32066)

*Dm* CYRI

**A'** *Dm* CYRI vs *Hs* CYRI-A

*Dm* CYRI
*Hs* CYRI-A

**A''** *Dm* CYRI vs *Hs* CYRI-B

*Dm* CYRI
*Hs* CYRI-B

**B**

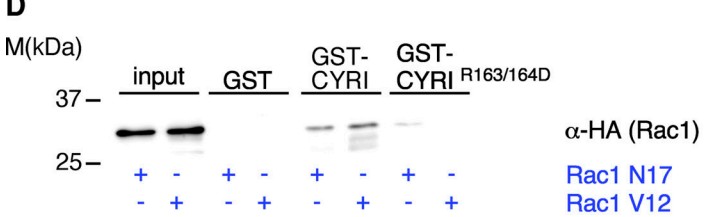

```
DmCYRI    DFSYYRRIVSRQR - 170
HsCYRI-A  DFSYYRRTISRNR - 167
HsCYRI-B  DFSYYRRZLSRMR - 167
Identity  ******* :** *
```

**C**

|  | *Dm*CYRI | *Hs*CYRI-A | *Hs*CYRI-B |
|---|---|---|---|
| *Dm*CYRI | 100 | 52.52 | 55.17 |
| *Hs*CYRI-A | 52.52 | 100 | 79.57 |
| *Hs*CYRI-B | 55.17 | 79.57 | 100 |

**D**

M(kDa)

| input | GST | GST-CYRI | GST-CYRI R163/164D |

37 –

25 –

α-HA (Rac1)

Rac1 N17 + - + - + - + -
Rac1 V12 - + - + - + - +

**E**

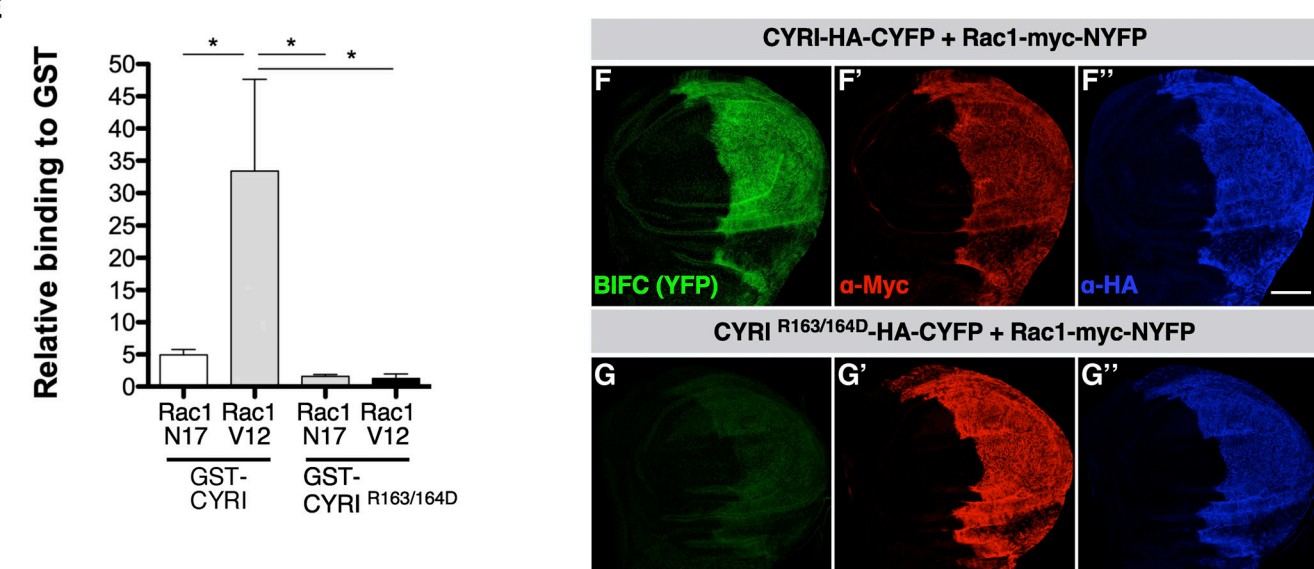

Relative binding to GST

Rac1 N17 | Rac1 V12 | Rac1 N17 | Rac1 V12
GST-CYRI | GST-CYRI R163/164D

CYRI-HA-CYFP + Rac1-myc-NYFP

**F** BIFC (YFP) **F'** α-Myc **F''** α-HA

CYRI R163/164D-HA-CYFP + Rac1-myc-NYFP

**G** BIFC (YFP) **G'** α-Myc **G''** α-HA

Figure 1. **Drosophila CYRI binds activated Rac1 and controls lamellipodial protrusions. (A–A'')** Comparisons of the highly similar structures and topologies of (A) CG32066 with human (A') CYRI-A and (A'') CYRI-B based on AlphaFold2 protein structure predictions (Jumper et al., 2021). As recently shown by crystal

structure analysis CYRI proteins comprised solely of α-helices (Kaplan et al., 2020; Yelland et al., 2021). Note the model contains the two highly conserved arginines at positions 163 and 164 (marked in yellow) in the fly protein corresponding to R161 and R162 in human CYRI-B. **(B)** Sequence alignment of *Drosophila* CG32066, human CYRI-A, and CYRI-B shows conservation of the arginines at positions 163 and 164. **(C)** Comparative sequence analysis between CG32066 and human CYRI-A and CYRI-B proteins. The numbers refer to Clustal W sequence alignment score (Thompson et al., 1994). **(D)** Pull-down experiments with GST-CYRI proteins. GSH-sepharose-bound GST-CYRI (wild type or R163/164D mutant) were preloaded with GDP or GTPγS and incubated with lysate from S2R+ cells transfected with either constitutively activated Rac1-V12 or dominant-negative Rac1-N17 construct. **(E)** Quantification of (D) from three independent experiments. Signals were normalized to GST. Mean ± SD. Statistical analysis using one-way ANOVA with Tukey's multiple comparisons. * P <0.05. **(F)** Visualization of CYRI and Rac1 BIFC interaction in wing imaginal discs. Maximum intensity projection images of wing imaginal discs expressing the indicated Split-YFP construct combinations in the *en*-Gal4 pattern. Expression of transgenes is verified by antibody staining as indicated. Anterior is to the left. **(G)** Co-expression of Rac1-myc-NYFP and wild type CYRI-HA-CYFP leads to reconstitution of YFP, whereas (G) co-expression of Rac1-myc-NYFP and mutant CYRI-R163/164D-HA-CYFP does not show YFP fluorescence. Three independent experiments for each genotype were performed. Scale bars represent 50 μm. Source data are available for this figure: SourceData F1.

---

advantage of CRISPR/Cas9-mediated genome editing to introduce small deletions within the first and third exon of the *cyri* gene locus (Fig. 3 A). We isolated two frame-shift mutants, *cyri*Δ2 and *cyri*Δ11 (Fig. 3 A). The homozygous viable *cyri*Δ2 mutant contains a small 2-bp deletion in the third exon after codon 174 that produces a frameshift and consequently results in the incorporation of 34 ectopic amino acids followed by a premature stop codon (Fig. 3 A). In comparison, the *cyri*Δ11 allele carries an 11-bp deletion in the first exon leading to a non-functional peptide. In contrast to *cyri*Δ2, homozygous *cyri*Δ11 mutant animals die at the early second instar larval stage. This discrepancy prompted us to test trans-heterozygous flies with the sequence mapped null deficiency Df(3L)ED4457 that removes the complete *cyri* gene locus. Interestingly, *cyri*Δ11/Df(3L)ED4457 were viable suggesting that *cyri*Δ11 also contains off-target mutations associated with lethality. To test protein expression in homozygous and trans-heterozygous *cyri* mutants, we applied Western blot analysis from wildtype and mutant fly extracts. Western blots showed that the antibody recognized endogenous 36 kDa CYRI protein in wildtype (Fig. 3 B). By contrast, extracts from homozygous and trans-heterozygous *cyri* mutant flies showed a loss of CYRI protein (Fig. 3 B). Of note, no truncated CYRI protein (~23 kDa in size) is detectable in *cyri*Δ2 mutant lysates (Fig. 3 B). Thus, we decided to further characterize both *cyri* alleles functionally.

In mammals, CYRI null cells exhibit a pancake-like morphology with extensive membrane ruffling and increased cellular spread (Fort et al., 2018; Whitelaw et al., 2019; Yuki et al., 2019). To analyze the effect of *Drosophila* CYRI on cell spreading, we isolated macrophages from homozygous and trans-heterozygous *cyri* mutant pupae (Fig. S1, E–H). We found that not only trans-heterozygous *cyri*Δ11 mutant (termed *cyri*Δ11/Df) but also *cyri*Δ2 mutant macrophages spread over a significantly larger area with unusually large and broad lamellipodia compared with wildtype (Fig. S1, E–H, quantification in Fig. S1 I). Compared with *cyri*Δ2, *cyri*Δ11/Df mutant macrophages showed the most prominent increased cell spread suggesting that *cyri*Δ2 is a hypomorphic allele rather than a null allele as *cyri*Δ11 (compare quantification in Fig. S1 I).

To further test a cell-autonomous function of CYRI in regulating actin-driven cell spreading, we expressed two different *cyri* RNAi transgenes under the control of macrophage-specific Gal4 driver (*hmlP2A*-Gal4; Stephenson et al., 2022). Expression of both *cyri* RNAi transgenes resulted in increased cell spreading

(Fig. 3, D and F). Notably, macrophages depleted for CYRI showed an increased immunofluorescent anti-WAVE intensity at the leading edge compared with wildtype control cells (Fig. 3, D and G) whereas overexpression of CYRI induced a spiky morphology that resulted in a marked reduction of endogenous WAVE at lamellipodial tips (Fig. 3 E, quantification in Fig. 3 G). Therefore, like their mammalian counterparts, *Drosophila* CYRI opposes active WAVE complex recruitment to the plasma membrane.

### Loss of *cyri* accelerates epidermal wound closure

We next analyzed *Drosophila* CYRI function in vivo. We first focused on its possible role in epidermal wound closure, a physiological process that highly depends on dynamic actin-based protrusions and migration of epithelial cells (Rothenberg and Fernandez-Gonzalez, 2019; Tsai et al., 2018). We have recently established a single-cell wounding model system using epidermal cells from the dorsal side of the abdomen of early pupal stages (Fig. 4, A and A'; Lehne et al., 2022). To visualize these cells, we expressed a Lifeact-EGFP transgene under the control of the epidermis-specific A58-Gal4 driver (Galko and Krasnow, 2004). Wildtype cells respond to laser-induced cell ablation by the formation of broad lamellipodial protrusions at the wound edge within the first 3 min (Fig. 4 B). Lamellipodia decrease over time and coexist with an increasing number of contractile actin bundles, which contract laterally to pull cells forward and further contribute to wound closure (Fig. 4 B and Video 3). Lamellipodia formation of cells adjacent to the wound edge highly depends on the Rac-WRC-Arp2/3 actin machinery. Supporting this notion, RNAi-mediated knockdown of either WAVE, Arp2, or Arp3 strongly disrupted lamellipodia formation (Fig. 4, C–E and Video 3). Consistently, animals depleted for *wave*, *arp2*, and *arp3* showed not only strong defects in lamellipodia size within the first minutes upon wounding (Fig. 4 F) but also showed a delay in wound healing within the first 20 min (Fig. 4 F'). Later on, purse-string contraction of an actomyosin ring seems to compensate for initial lamellipodial cell migration defects in *wave*, *arp2*, and *arp3*-depleted epidermis. Wound measurements 60 min after injury revealed no significant difference to wildtype tissue (Fig. 4 F').

Strikingly, similar wound closure defects were observed in animals overexpressing a wildtype full-length CYRI but not a mutant CYRI-R163/164D transgene deficient for Rac binding (Fig. 5, A and B; quantification in Fig. 5, E and E'). To analyze wound

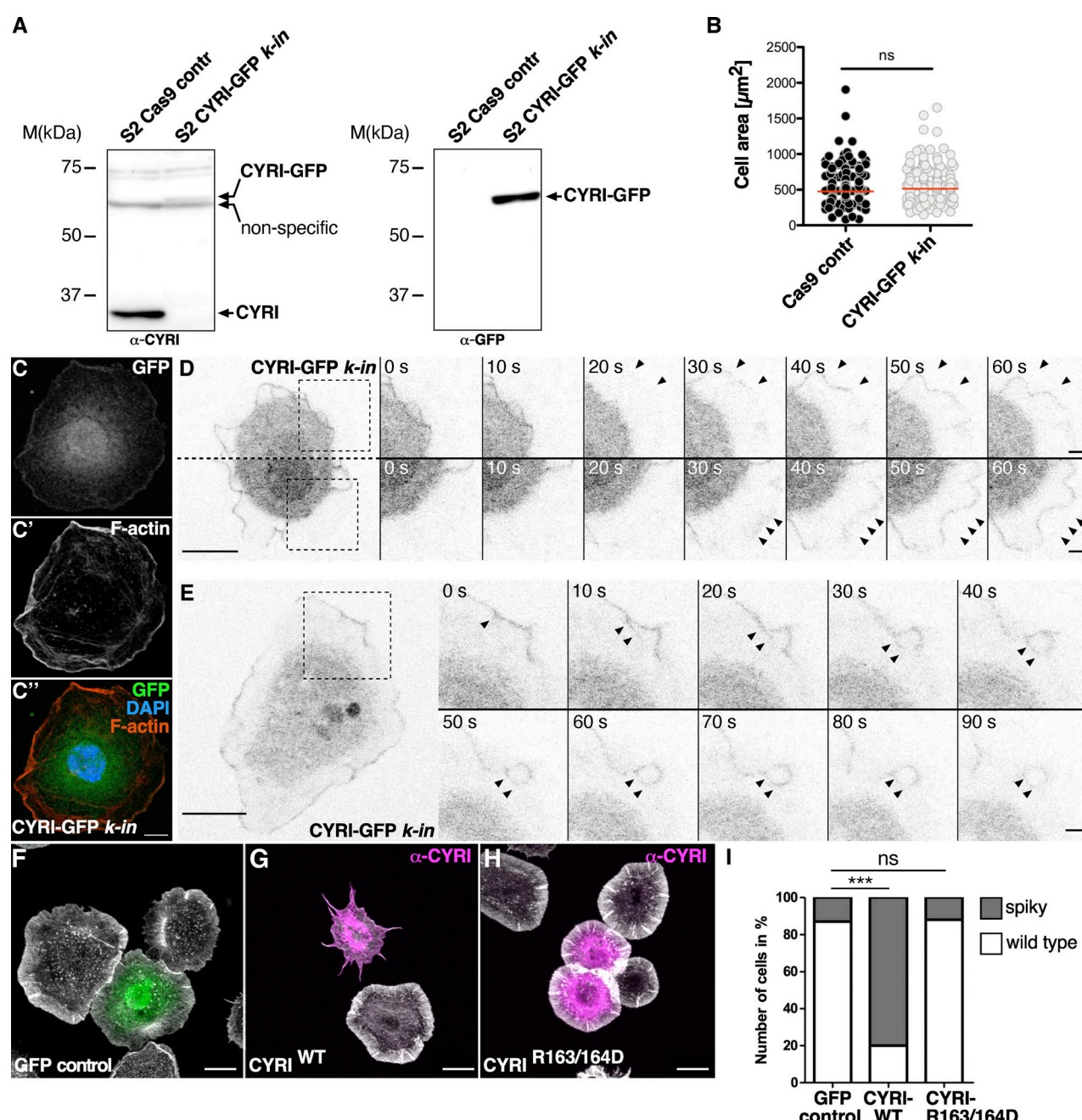

Figure 2. **Drosophila CYRI controls lamellipodial protrusions. (A)** Insertion of a GFP-tag at the 3′ end of the *cyri* gene in S2 cells using CRISPR/Cas9-mediated genome editing. Lysates from control cells and cells expressing CYRI-GFP were probed on a Western blot using an anti-CYRI (left) or anti-GFP (right) antibody. **(B)** Quantification of spread cell area, Cas9 S2 control: *n* = 200 cells; S2 CYRI-GFP knock-*in*: *n* = 200 cells; To evaluate statistical significance, the Mann–Whitney test was used and the following P value (two- tailed) was obtained: P value: ns < 0.05. **(C–C″)** Confocal images of CRISPR/Cas9-edited S2 cells expressing CYRI-GFP (C endogenous GFP; green) stained with phalloidin-Alexa568 (C′ red) and DAPI (C″ blue). Scale bars represent 10 μm. **(D and E)** Time-lapse fluorescence microscopy images of S2 CYRI-GFP knock-*in* cells. Images were taken at indicated timepoints. Black arrowheads mark lamellipodial protrusions in D and macropinocytic structures in E enriched for endogenous CYRI-GFP. Scale bar represents 10 μm. **(F–H)** Confocal images of S2R+ cells stained with phalloidin-Alexa488 (grey) and an anti-CYRI antibody (magenta) transfected with (F) an EGFP, (G) a wild type CYRI (CYRI^WT) or (H) a mutant CYRI-^R163/164D construct. **(I)** Quantification of cells showing a spiky cell morphology. *n* = 100 cells for each genotype from three independent transfection experiments. Two-sided Fisher's exact test was used. P value: *** P < 0.0001; ns: > 0.05. Source data are available for this figure: SourceData F2.

closure in *cyri* mutant animals, we recombined the A58-Gal4 driver and the UAS-Lifeact-EGFP reporter on the mutant chromosomes. Interestingly, loss of *cyri* function increased lamellipodia size and promoted wound closure (Fig. 5, C and D; quantification in Fig. 5, E and E′; and Video 4). The strongest

positive effect on wound healing was observed in *cyri*Δ11 mutant epidermis compared with hypomorphic *cyri*Δ2 (Fig. 5, E and E′). Remarkably, mutant wounds almost completely closed after 1 h, a healing efficiency that was rarely observed in wild type epidermis (Fig. 5 E). Notably, re-expression of a wild type CYRI but

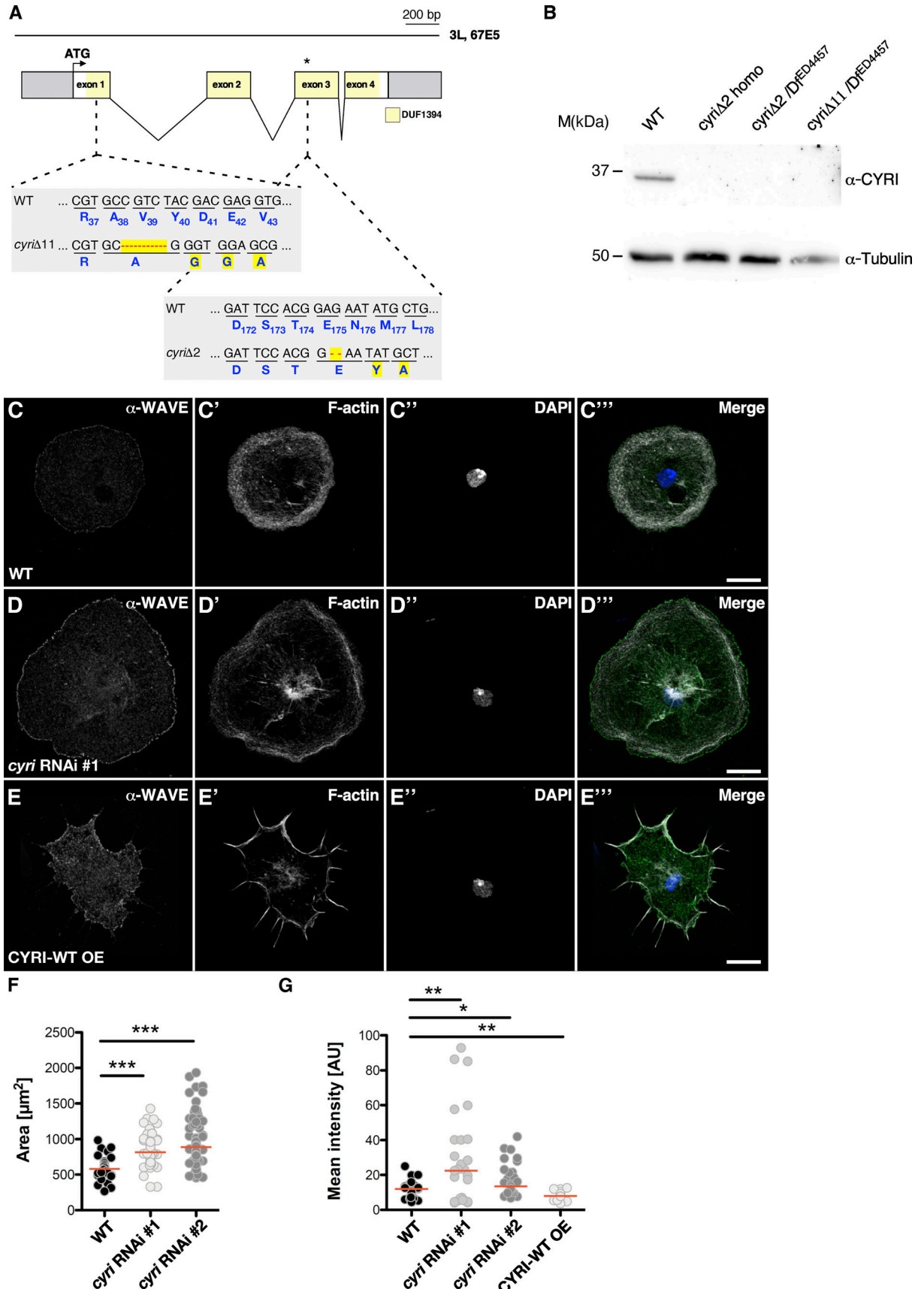

Figure 3. **Loss of CYRI controls lamellipodia spread of macrophages. (A)** Schematic overview of the *cyri* gene locus. Exons encoding parts of the DUF1394 domain are highlighted in yellow. The target sequence for CRISPR/Cas9 gene modification and generated *cyri* deletions, *cyri*Δ2 and *cyri*Δ11 is depicted. **(B)** Loss

of *cyri* mutants were validated by western blot analysis using a specific anti-CYRI antibody. Lysates from ten ovaries of wild type and different mutant flies were analyzed. Anti-tubulin signal served as a loading control. **(C–E)** Confocal images of (C) wild type; (D) *cyri* RNAi depleted (E) CYRI-WT overexpressing macrophages stained for endogenous WAVE (α-WAVE; green), F-actin (grey), and DAPI (blue). Scale bars represent 10 µm. **(F)** Quantification of spread cell area of pre-pupal hemocytes (wild type: 30 cells; *cyri* RNAi #1: 89 cells; *cyri* RNAi #2: 85 cells.). Statistical significance was evaluated using one-way-ANOVA (Kruskal–Wallis test) followed by Dunn's Multiple Comparison test. P value: <0.0001 (***). The red bar represents the median. Three independent transfection experiments were performed. **(G)** Quantification of immunofluorescent anti-WAVE intensity at the leading edge of wild type (n = 30 cells), *cyri* RNAi depleted macrophages (two different RNAi transgenes #1 and #2; each n = 29 cells) and macrophages overexpressing a wild type CYRI transgenes (CYRI-WT OE, n = 30 cells) normalized to background fluorescence. One-way-ANOVA test was performed. For multiple comparison, the test was corrected after Dunnett. P = *(0.033), **(0.002), ***(0.001). Quantification was done from three independent experiments. Source data are available for this figure: SourceData F3.

not Rac-binding deficient CYRIR163/164D variant rescued the *cyri* loss of function phenotype resulting in wound closure behavior similar to wild type epidermal cells (Fig. 5 E' and Video 5). Increased wound-induced lamellipodia formation observed in *cyri*-depleted cells indeed seems to be due to an increased Rac activity. We established a new GFP-based sensor for active Rac1 and Rac2, termed MBT-GFP. This sensor includes the Cdc42- and Rac-interactive binding (CRIB) domain of the p21-activated kinase mushroom bodies tiny (Mbt) fused to EGFP (Melzig et al., 1998). GST-pull down experiments confirmed its specific binding to GTP-loaded *Drosophila* Rac1 and Rac2 (Fig. S2, A and B). MBT-GFP also strongly bound Cdc42, but this binding was, however, independent of the nucleotide-binding status of Cdc42 (Fig. S2, A and B). In vivo, the reporter nicely marks increased lamellipodial protrusions at the wound margin of epithelial cells depleted for *cyri* (Fig. 5, F and G; and Video 6). Quantification confirmed both significantly increased mean and maximum fluorescence intensity upon *cyri* knockdown (Fig. 5 H).

In addition, we performed similar localization experiments with an Abi-EGFP transgene, which has been shown to resemble WRC localization in vivo (Squarr et al., 2016, *JCB*). Different from the MBT-CRIB-GFP sensor, Abi-EGFP exclusively marks the tips of protruding lamellipodia formed at the wound margin (Fig. 5 I and Video 7). Compared with wild type, RNAi-mediated suppression of CYRI resulted in a significantly increased maximum intensity of Abi-EGFP localization at lamellipodial tips (Fig. 5 J, quantification in Fig. 5 K and Video 7). Thus, these additional data further support that CYRI acts as a molecular brake on the Rac-WRC-Arp2/3 pathway in wound healing by opposing active WAVE complex recruitment at the wound edge.

To further validate the cell-autonomous function of CYRI in epidermal wound healing, we expressed the two different *cyri* RNAi transgenes under the control of the epidermis-specific A58-Gal4 driver. Both RNAi fly lines phenocopied *cyri* loss-of-function and resulted in a significantly increased lamellipodia formation in cells around the wound edge (Fig. S3 A). Loss of CYRI function accelerates wound closure not only in single-cell wounds but also in multicellular epidermal wounds (four to six cell ablation), significantly healing faster in cells lacking CYRI function (Fig. S3, B–D). Interestingly, simultaneous RNAi-mediated suppression of *cyri* and *wave* still resulted in lamellipodial protrusion defects similar to *wave* RNAi depletion alone (Fig. S3 A). Thus, this further suggests that CYRI indeed acts through WAVE in regulating lamellipodial protrusions during wound closure.

## Loss of *cyri* function affects border cell cohesion and cluster migration

While trans-heterozygous *cyri*Δ11 mutants are viable, we observed reduced female fertility resulting in significantly lesser offspring (Fig. 6 A). Reduced fertility correlated with defects in the formation of the micropyle, the structure through which the sperm enters to fertilize the egg (Montell et al., 1992). Examination of the eggs from *cyri*Δ11 mutant females revealed that many of them showed a shortened micropyle compared with the wild type (Fig. 6, B and C). Micropyle formation is closely linked to proper border cell migration during egg development (Horne-Badovinac, 2020). Border cells form a small group of 6–10 somatic cells that delaminate from the follicle epithelium and migrate invasively and collectively between the nurse cells toward the border between the oocyte and nurse cells (Montell et al., 2012; Peercy and Starz-Gaiano, 2020). Thus, we suspected the possibility that *cyri* mutation may impair collective border cell migration and looked more closely into the cellular basis of reduced egg fertility.

Recent studies by the Montell lab support an integrated model of collective border cell migration in which the highest Rac activity is found in the leader cell which steers the cluster, but Rac is also required in the follower cells that coordinate both cluster migration and cluster cohesion (Campanale et al., 2022; Wang et al., 2010). The border cell cluster contains two cell types: a pair of non-motile cells, so-called polar cells, which initiate the cluster delamination; and four to eight outer, motile border cells, which carry the polar cells to the border between the oocyte and nurse cells (Fig. 6 D; Montell et al., 1992). Wild type border cells are specified in stage 8, migrate invasively in between the nurse cells during stage 9, and arrive at the nurse cell-oocyte boundary by stage 10 (Fig. 6, D and E). We indeed observed striking defects in *cyri*Δ11 mutant stage 9 and 10 egg chambers when we followed the outer migratory border cells stained with an antibody against nuclear Eyes Absent (EYA) (Bai and Montell, 2002). In *cyri*Δ11 mutant egg chambers, border cells started migration but single border cells often detached from and trailed behind the main cluster to ultimately remain between the nurse cells (Fig. 6, F and G; magnifications in Fig. 6, F1 and G1–G3; quantification in Fig. 6 H). As a consequence, mutant border cell clusters that reached the nurse cell–oocyte boundary often contained less migratory border cells (Fig. 6 G3). The total cell number of the border cell cluster, however, was not changed (Fig. S4 A).

We further validated the cell-autonomous function of CYRI in border cell cohesion by cell-type specific RNAi experiments. RNAi-mediated depletion in outer border cells using the c306-

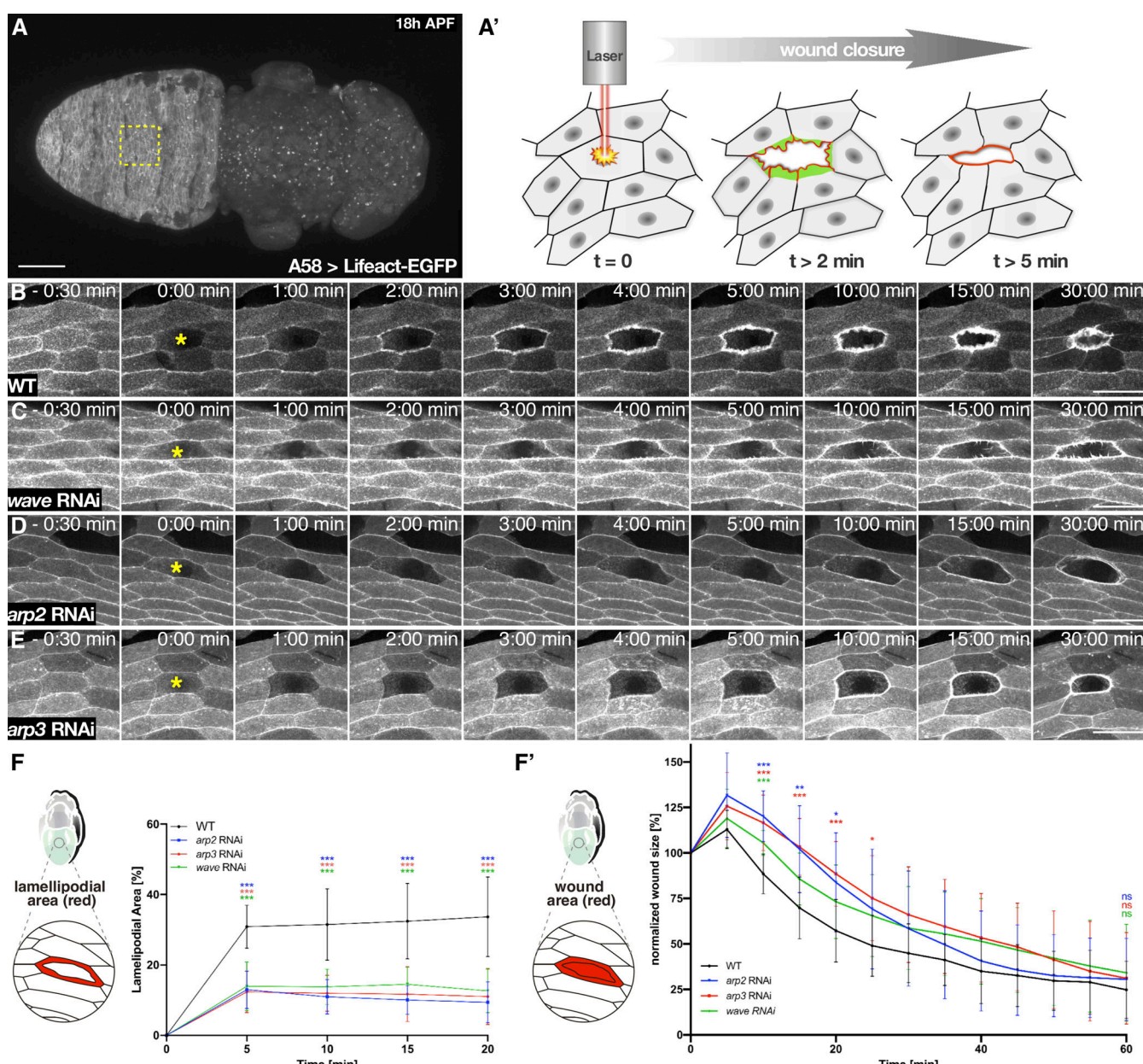

Figure 4. **The Rac-WRC-Arp2/3 pathway is required for wound closure. (A)** Wild type 18 h APF old pupa specifically expressing a Lifeact-EGFP transgene in the abdominal epidermis under the control of the *A58*-Gal4 driver. The imaged area of the monolayered epithelium is boxed in yellow. **(A')** Scale bar represents 250 µm (A') Schematic of the in vivo wounding model. Laser-induced single-cell ablation starts at $t = 0$ min. In the first two min ($t = 2$ min), F-actin assembles into broad lamellipodial protrusions (green) within cells at the wound edge; lamellipodial protrusions reach a maximum size between 5 and 10 min after wounding. Later on ($t > 5$ min), an acto-myosin ring (red) is formed at the leading edge of the wound (according to Lehne and Bogdan [2023]). **(B–E)** Frames of spinning disc microscopy videos of 18 h APF old epidermis expressing a Lifeact-EGFP transgene under the control of the *A58*-Gal4 driver. The genotypes are indicated. Images were taken at indicated time points. Ablation of a single cell (yellow asterisk) starts at $t = 0$ min. Scale bar represents 50 µm. **(F and F')** Quantification of wound closure in wild type (WT; $n = 13$) and after knockdown of *arp2* ($n = 15$), *arp3* ($n = 16$) and *wave* ($n = 16$) by RNAi. **(F)** Lamellipodia size was measured every 5 min and normalized to the initial size of the unwounded cell. Scale bar represents 50 µm. **(F')** After 60 min wound closure was assessed by comparison of remaining wound size normalized to unwounded cell size. To evaluate statistical significance in F and F', the two-way ANOVA analysis with Dunnett correction was used and P values was obtained: P value: 0.12 (ns), 0.033 (*), 0.002 (**), <0.001 (***). Error bars represent SD. At least three independent experiments for each genotype were performed.

Gal4 driver line also resulted in prominent lagging border cells (Fig. 7 A). Expression of the same two RNAi transgenes under the control of the *upd*-Gal4 driver (E132-Gal4), which is exclusively expressed in polar cells, did not result in any significant phenotype, suggesting that CYRI function is only needed in migratory outer border cells (Fig. 7 A'). Interestingly, overexpression of wild type CYRI but not Rac-binding deficient CYRI[R163/164D] variant under the control of the c306-Gal4 driver did not result in migration defects of border cells clusters (Fig. 7 C; quantification in Fig. 7 A). Both, delayed migration and

Figure 5. **Loss of CYRI accelerates epidermal wound closure. (A and B)** Single-cell ablation experiments in the abdominal epidermis overexpressing a (A) wild type CYRI and (B) mutant CYRI[R163/164D] transgene. **(C and D)** Single-cell ablation in the abdominal epidermis of homozygous *cyriΔ2* mutant, (D) and a transheterozygous *cyriΔ11/Df(ED4457)* mutant. Images were taken at indicated time points. Ablation of a single cell (yellow asterisk) starts at *t* = 0 min. Scale bars represent 50 µm. **(E and E')** Quantification of wound closure defects. Following genotypes were measured: wild type (WT; *n* = 13), cyriΔ2 (*n* = 14), cyriΔ11/Df(ED4457) (*n* = 15), CYRI[WT] (*n* = 15), CYRI[R163/164D] (*n* = 18), rescue with CYRI[WT] (*n* = 7), and rescue with CYRI[R163/164D] (*n* = 11). **(E)** Over the time of 60 min wound closure was assessed by comparing the remaining wound size normalized to the unwounded cell size. **(E')** Lamellipodia size was measured every 5 min and

normalized to the initial size of the unwounded cell. To evaluate statistical significance in E and F, the two-way ANOVA analysis with Dunnett correction was used and the following P values were obtained: P value: 0.12 (ns), 0.033 (*), 0.002 (**), <0.001 (***). Error bars represent SD. At least three independent experiments for each genotype were performed. Note that wild type controls are the same as in Fig. 4, F and G as the data belong to the same dataset. Note that wounded transheterozygous *cyri*Δ11 mutant epithelium almost completely sealed after 40 min. **(F and G)** Frames of spinning disc microscopy videos of 18 h APF old epidermis expressing the Rac sensor MBT-GFP under the control of the A58-Gal4 driver. The genotypes are (F) wild type and (G) *cyri* RNAi. Images were taken at indicated time points. Ablation of a single cell (yellow asterisk) starts at *t* = 0 min. Red arrowheads mark increased broad lamellipodia marked by MBT-GFP in epidermal cells at the wound margin depleted for *cyri*. Scale bar represents 20 μm. **(H)** Quantification of the mean and maximum fluorescence intensity of MBT-GFP in control and *cyri* RNAi–depleted epidermis. Statistical significance was determined by Unpaired *t* test with Welch's correction, P value: <0.001 (***) (wild type: *n* = 10; *cyri* RNAi: *n* = 10). At least three independent experiments for each genotype were performed. **(I and J)** Frames of spinning disc microscopy videos of 18 h APF old epidermis expressing the WRC subunit Abi-GFP under the control of the A58-Gal4 driver. The genotypes are (I) wild type and (J) *cyri* RNAi. Images were taken at indicated timepoints. Ablation of a single cell (yellow asterisk) starts at *t* = 0 min. Red arrowheads mark increased broad lamellipodial tips marked by Abi-GFP in epidermal cells at the wound margin depleted for *cyri*. Scale bar represents 20 μm. **(K)** Quantification of the maximum fluorescence intensity of Abi-GFP in control and *cyri* RNAi depleted epidermis. Statistical significance was determined by Unpaired *t* test with Welch's correction, P value <0.001 (***) (wild type: *n* = 12; cyri RNAi: *n* = 12). At least three independent experiments for each genotype were performed.

cohesion defects of border cell cluster were also seen upon RNAi-mediated depletion of WAVE (Fig. 7 D; quantification in Fig. 7 A). Given that Rac activity must be tightly regulated in both, leader and follower cells, increased pools of activated Rac might affect not only migration but also the cohesion of the border cell cluster (Campanale et al., 2022). Supporting this notion, we found that overexpression of a membrane-tethered activated WAVE variant (WAVE^Myr; Stephan et al., 2011) in outer border cells, phenocopied loss of *cyri* function, thus resulting in a significant reduction of cluster cohesion (Fig. S4, B–D; quantification in Fig. 7 A). Notably, the number of WAVE^Myr expressing egg chambers with prominent lagging border cells was lower compared to *cyri*Δ11 or cyri RNAi (compare Fig. 7 A), suggesting that increased Rac-dependent WRC activation cannot fully account for cluster cohesion defects in *cyri* deficient border cell clusters. Double RNAi experiments further indicated no simple epistatic relationship between *cyri* and *wave* in border cell cohesion different from wound closure. Instead, we observed a significant decrease of lagging border cells when we depleted both *cyri* and *wave* compared with single RNAi depletion (Fig. 7 A). Removal of one copy of *wave* in *cyri* mutant background did not significantly reduce the lagging border phenotype (Fig. S4 E). For this reason, we further analyzed whether CYRI might directly affect border cell cluster cohesion.

A central adhesion molecule is E-cadherin which mediates adhesion between border cells and nurse cells (Cai et al., 2014; Niewiadomska et al., 1999). Loss of E-cadherin in either cell type blocks migration and results in elongated clusters with single cells trailing behind (Niewiadomska et al., 1999). In wild type, the highest concentration of E-cadherin is found at the apical interface between border cells and polar cells (apical cap, ring-like structure) and at the contact side between adjacent border cells (BC-BC interface, "arms"; Fig. 8 A', B, and D; magnification in B''', D'''). At the interface between border cells and nurse cells substantially lower amounts of E-cadherin are detectable. Migrating border cell clusters maintain the apico-basal polarity, and the apical cap/ring structure is oriented approximately orthogonal to the direction of migration (Fig. 8 A''; Felix et al., 2015; Niewiadomska et al., 1999; Pinheiro and Montell, 2004). We found that E-cadherin localization and levels were not changed in *cyri*Δ11 mutant egg chambers (Fig. 8, C and E; magnification in C''', E'''), suggesting that cluster cohesion defects are not due to changes in E-cadherin-mediated adhesion.

Instead, we observed changes in the localization of βPS-integrin, the second adhesion molecule that contributes more to cluster cohesion than to cell–matrix interactions during border cell migration (Dinkins et al., 2008; Llense and Martín-Blanco, 2008). Expression of βPS-integrin, which marks the basal domain of the cluster, predominantly localizes at the interface between border cells (BC-BC) in wild type (Fig. 8, B and D; magnification in B''''' and D'''''). Unlike wild type, *cyri*Δ11 mutant clusters showed a changed localization along the basal-lateral domain (Fig. 8, C and E, magnification in C''''', E'''''). While E-cadherin was still enriched in the apical domain, less βPS-integrin localized to the basal side and along the lateral interface between *cyri*Δ11 mutant epithelial cells but showed reduced βPS-integrin membrane localization compared with wild type clusters (Fig. 8 F; see also Video 8). By contrast, overexpression of a wild type CYRI transgene under the control of the c306-Gal4 driver line did not result in significant changes in βPS-integrin localization (Fig. 8 G). Taken together, these data suggest that changed integrin-mediated adhesion between border cells might also contribute to cohesion defects in *cyri* mutants.

## Discussion

CYRI proteins have been previously identified as a new class of Rac1 interactors that interfere with WRC activation (Fort et al., 2018; Whitelaw et al., 2019; Yuki et al., 2019). At the single cellular level, these studies revealed an important role of CYRI proteins in regulating cell migration, macropinocytosis, and pathogen entry into cells. In this study, we have used *Drosophila* as a model system to study CYRI function at cellular and organismal levels. Our data confirmed an evolutionary conserved function of CYRI controlling lamellipodia spread and protrusion dynamics by opposing Rac-mediated activation of WRC. CYRI is not only a potent and dynamic regulator of WRC in single macrophages but also in epithelial tissue wound healing. Our data highlight a novel role of CYRI in Rac-WRC-Arp2/3-dependent epidermal wound closure. Laser-induced wounding experiments suggest that CYRI acts as a molecular brake on the Rac-WRC-Arp2/3 pathway to slow down wound healing possibly to enable an efficient inflammatory response and improve proper re-epithelization and scarring. Supporting this notion, loss of CYRI function accelerates wound healing, whereas its

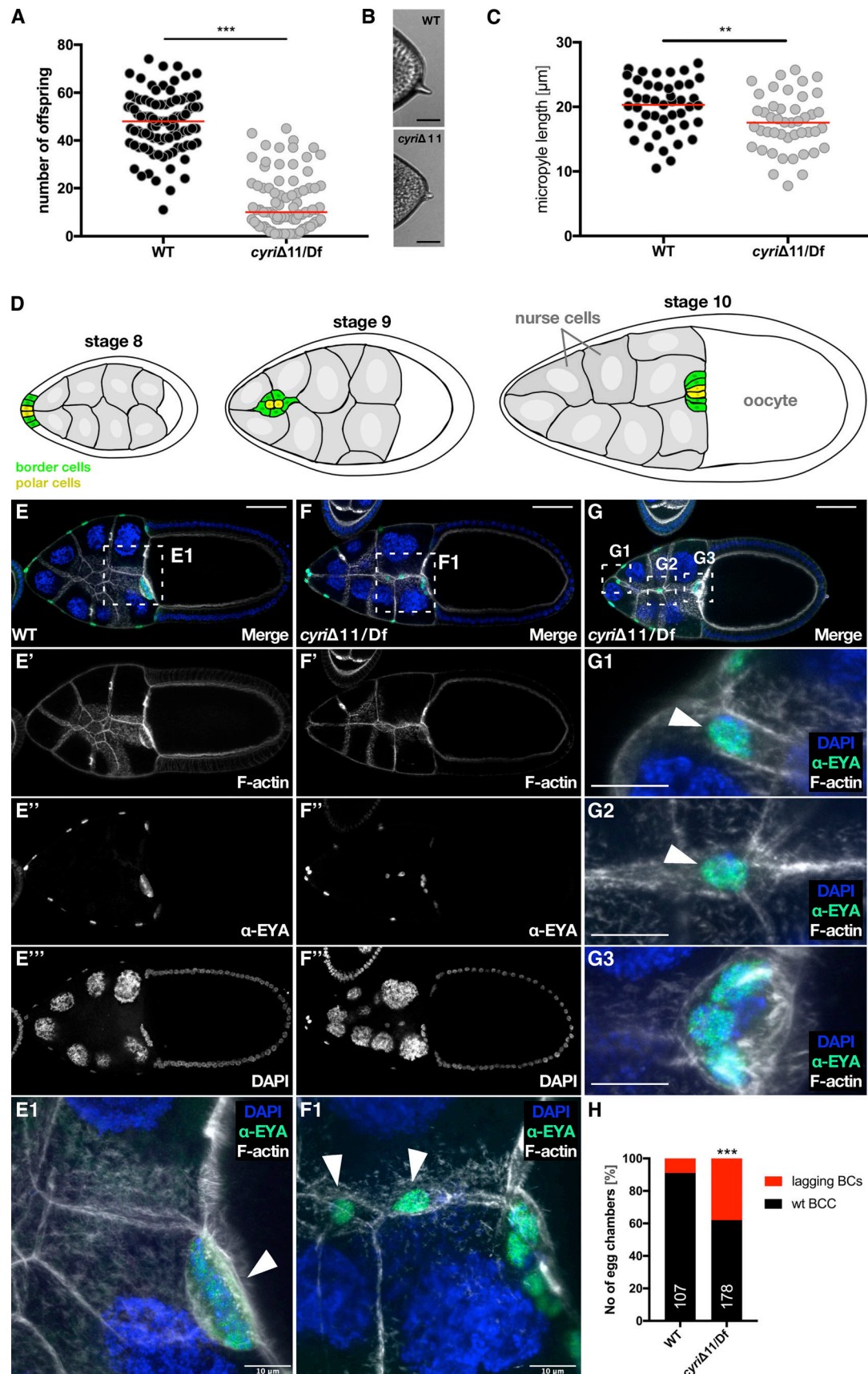

Figure 6. **Loss of CYRI results in partial sterility and border cell migration defects. (A)** Quantification of *cyri* mutant female fertility. Single mutant females were mated with wild type males and the total number of offspring reaching adulthood was counted. Transheterozygous mutant cyriΔ11/Df(ED4457)

females had substantially reduced fertility and produced fewer offspring compared with wild type. (n = 95; wild type and n = 94; cyriΔ11/Df(ED4457); the red bar represents the median. Mann–Whitney test was used to determine statistical significance: P value<0.001. **(B)** Brightfield photomicrographs of wild type and *cyri* mutant eggs. **(C)** Quantification of micropyle length. (n = 45; wild type and n = 45; cyriΔ11/Df[ED4457]); The red bar represents the median. Mann–Whitney test was used to determine statistical significance: P value: 0.004. Three independent experiments were performed. **(D)** Schematic drawing of border cell migration during egg development. Polar cells are marked in yellow and border cells in green. **(E–G)** Maximum intensity projections of three confocal slices of stage 10 egg chambers of the indicated genotypes with DNA (DAPI, blue), F-actin (grey) and anti-EYA (green); anterior is to the left. **(E)** Wild type egg chamber (F and G) two examples of *cyri* mutant egg chambers showing prominent lagging border cells. Bars represent 50 μm. **(E1)** Detailed view of boxed area in E showing wild type border cell cluster arrived the nurse cell-oocyte border. **(F1)** Detailed view of the boxed area in (F) shows an abnormally elongated border cell cluster with some cells completely detached. **(G1–G3)** High magnification of boxed areas in G shows cells detached from the main border cell cluster. The scale bar represents 10 μm. **(H)** Quantification of border cell cluster of indicated genotype with lagging border cells. Statistical significance was determined by Fisher's exact test, P value <0.001 (wild type: n = 107; cyriΔ11/Df[ED4457]: n = 178). At least three independent experiments for each genotype were performed.

---

overexpression suppresses epidermal re-epithelialization. High-resolution live imaging of wounded mutant tissue showed dramatically increased lamellipodia protrusions that assembled in cells around the wound edge and contributed together with acto-myosin-based contraction to efficient wound closure. This suggests that the WRC-Arp2/3 complexes are hyperactivated by Rac in *cyri*-deficient epithelial cells.

How could CYRI oppose Rac1-mediated activation of WRC? CYRI might prevent the formation of clusters or dimers of WRC by limiting the amount of available GTP-bound Rac1 (Machesky, 2023). Such hetero- and homo-oligomeric complexes of WRC have been previously observed at integrin junctions in wing epithelia (Gohl et al., 2010). However, we found no evidence for such a possible scenario in gel filtration chromatography experiments (Fig. S4 F). By contrast, endogenous WRC protein complexes from lysates of *Drosophila* S2 cell overexpressing either wild type CYRI or the Rac-binding deficient CYRI$^{R163/164D}$ variant were still cofractionated with similar high molecular weight complexes at 500–700 kDa (Fig. S4 F). CYRI might either compete with WRC and/or Rac GTP exchange factor (GEF) proteins to limit the amount of active Rac1 (Machesky, 2023). The conserved Myoblast city (Mbc)-ELMO/CED-12 complex is known to act as a member of Rac GEFs to control Rac1 activity and lamellipodia formation in *Drosophila* dorsal closure, a morphogenetic movement of two opposing epithelial sheets similar to wound healing (Toret et al., 2018). However, the role of Mbc-Elmo in epidermal wound healing has not yet been addressed. Biochemical studies showed that the WRC requires two active Rac1 molecules simultaneously and CYRI acts by specifically disrupting Rac1–WRC interactions using its A-site-analogous DUF 1394 domain (Ding et al., 2022). Thus, if CYRI removes one of these, it acts as a potent and dynamic regulator of WRC. Given the prominent enrichment of endogenously GFP-tagged *Drosophila* CYRI at the leading edge of cultured S2 cells CYRI behaves like a "local inhibitor" of WRC, sequestering Rac away from interaction with WRC at the cell membrane as previously suggested (Fort et al., 2018).

CYRI function is not limited to the re-epithelialization dynamics in the epidermis but also in other epithelia during tissue morphogenesis that also require a tightly regulated collective cell behavior. Strikingly, we found that CYRI regulates the cohesion of border cell cluster, a well-established *Drosophila* model for studying invasive, collective cell migration in the physiological context of fly oogenesis. Collective cell migration requires efficient coordination of cell–cell and cell–matrix interactions. We found that loss of CYRI function results in a reduced cohesion of border cells, a phenotype reminiscent of defects in cell–cell and cell–matrix interactions. E-cadherin expression appears grossly normal in *cyri* mutant egg chambers, but mutant clusters showed a changed localization of β-integrin along the basal-lateral domain at border cell contacts. As a consequence, single border cells often trailed behind the main cluster to ultimately remain between the nurse cells, reminiscent of defects seen in βPS-integrin knockdown (Dinkins et al., 2008; Llense and Martín-Blanco, 2008). Interestingly, in cultured mammalian cells, depletion of both CYRI-A and CYRI-B resulted in enhanced surface expression of the α5β1 integrin via reduced internalization suggesting a conserved role of CYRI proteins in integrin trafficking (Le et al., 2021).

Defects in border cell cohesion partly explain the reduced fertility observed in *cyri* mutant females. Overexpression experiments further suggest that CYRI not only controls border cell cohesion but also WAVE-driven cluster motility. Supporting this notion, overexpression of CYRI phenocopies *wave* knockdown resulting in delayed migration of border cells. These data show that the level of CYRI expression is functionally important and must be tightly regulated in vivo. Expression data in different types of cancer suggest that human CYRI proteins also play an important pathophysiological function in invasive collective cell behavior of epithelial tissues (Nikolaou and Machesky, 2020). Most invasive solid tumors display predominantly collective invasion, in which groups of cells invade the peritumoral stroma while maintaining cell–cell contacts (Friedl et al., 2012; Friedl and Wolf, 2003). A reduction of cohesion of epithelial cells within the tumor or stromal cells is often accompanied by local or distant metastasis, which is also a hallmark of cancer. Interestingly, CYRI-B (FAM49B) has been originally identified as a suppressor of cancer cell proliferation and invasion in pancreatic ductal adenocarcinoma (PDAC; Chattaragada et al., 2018). Scratch-wound and matrigel invasion assays revealed that FAM49B/CYRI-B-silenced PDAC cells and CYRI-A/CYRI-B double knockout A-673 cells displayed enhanced migration compared with control cells (Chattaragada et al., 2018; Le et al., 2021). Since CYRI-B is also highly expressed in the human skin (Uhlén et al., 2015), it will be interesting to determine whether CYRI-B might also play an important role in skin wound healing and repair.

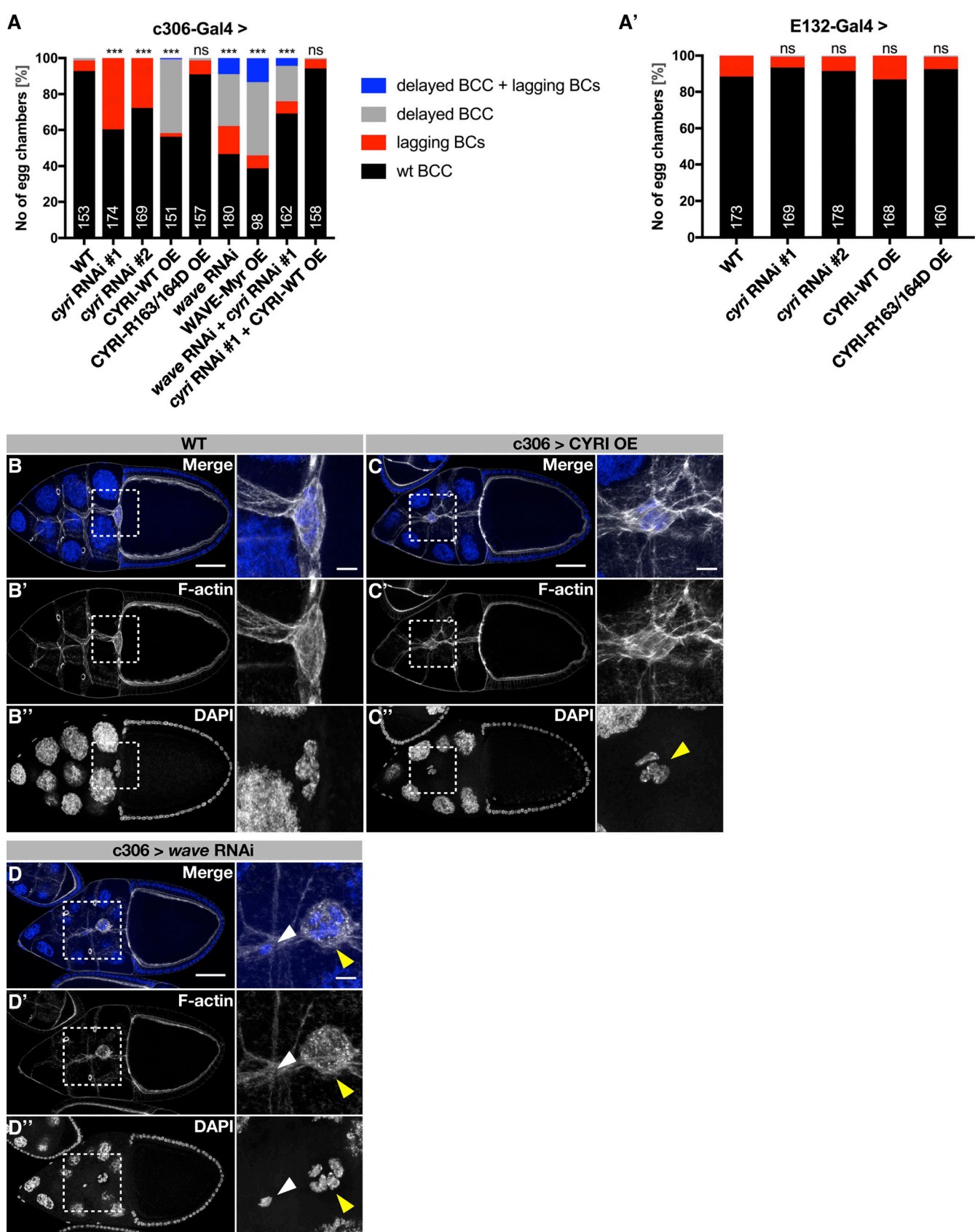

Figure 7. **CYRI controls border cell cluster cohesion and migration. (A and A')** Quantification of border cell cluster defects (lagging border cells and delayed border cell clusters) under the control of the (A') c306-Gal4 and (A') E132-Gal4 (*upd*-Gal4) driver. wild type; *cyri* RNAi #1; *cyri* RNAi #2; CYRI-WT OE; CYRI-R163/164D; *wave* RNAi and WAVE-Myr OE. N numbers are indicated. Quantified phenotypes are indicated by the colored legend in the middle. All

genotypes were compared against WT using the chi-square test, P values: ns > 0.12, ***<0.001. At least three independent experiments for each genotype were performed. **(B–D)** Maximum intensity projections of stage 10A egg chambers of the indicated genotypes with DNA (DAPI, blue) and F-actin (red); anterior is to the left. Detailed views of the boxed area are shown on the right. **(B)** wild type; (C) CYRI overexpression (OE), yellow arrowhead marks nuclei of delayed border cell cluster; (D) RNAi-mediated suppression of *wave* function, yellow arrowhead marks delayed border cell cluster, whereas white arrowhead marks a lagging border cell.

## Materials and methods

### *Drosophila* genetics

Flies were kept at room temperature in plastic vials, containing *Drosophila* standard food. For experimental procedures, fly husbandry and crossings were carried out according to standard methods and were kept at 25°C. In UAS-Gal4-based experiments, crossings were kept at 29°C. The following lines were obtained from Bloomington stock center: w[1118] (BL3605), hmlΔ-Gal4 (BL30139), en-Gal4 (BL30564), c306-Gal4 (BL3743), and UAS-Lifeact.GFP (BL57326). The A58-Gal4 driver was a kind gift from M. Leptin (Galko and Krasnow, 2004). Further lines were obtained from the Vienna *Drosophila* RNAi Center: arp2 RNAi (v29944), arp3 RNAi (v35258), rac1 RNAi (v49246), rac2 RNAi (v50350), cyri RNAi #1 (v107318), cyri RNAi #2 (v44825) and from NIG-FLY: wave RNAi (4636R-1). Additionally, the following transgenic lines were used: UAST-CYRI$^{WT}$, UAST-CYRI$^{R163/164D}$, UAST-CYRI-HA-CYFP, and UAST-dRac1-myc-NYFP. These lines were generated in accordance with the protocol for the germline-specific φ31-integrase system published by Bischof et al. (2007). The following lines were used for injections: y [1] M{vas-int.Dm}ZH2A w[*]; 579 M{3xP3-RFP.attP'}ZH-86Fb (BL24749) and y[1] M{vas-int.Dm}ZH2A w[*]; M{3xP3-580 RFP.attP'}ZH-68E (BL24485). The cyriΔ2 and cyriΔ11 mutants were generated by CRISPR/Cas9 of the following target sequence: cyriΔ2: 5′-CGGAGAATATGCTGGTCAGC-3′ and cyriΔ11: 5′-GGAGCGTGCCGTCTACGACG-3′.

### Purification of recombinant CYRI and antibody generation

Expression of GST-tagged full-length *Drosophila* CYRI (pGEX 6P1; Addgene) was induced in *ArcticExpress* (DE3) cells (Agilent Technologies) by 1 mM IPTG for 24 h at 10°C. After the incubation, cells were harvested and lysed by sonication in 1× PBS containing a protease inhibitor cocktail (Roche). The protein-containing solution was purified in accordance with instructions of the used GSTrap FF column and the ÄKTA laboratory-scale affinity chromatography system (GE Healthcare). Guinea pigs were immunized with purified proteins by Pineda Antikörper services (Pineda, Berlin). The rabbit His-CYRI antibody was generated by expression of 6x-His-tagged full-length *Drosophila* CYRI (pDEST17; Themo Fisher Scientific) in BL21-AI *E. coli* (Thermo Fisher Scientific). Protein expression was induced by arabinose for 4 h at 37°C. The recombinant protein was purified with Ni-NTA resin and sent for immunization of rabbits to the Pineda Antibody service (Pineda; Berlin). Affinity-purification of antibodies was performed with immobilized GST-CYRI loaded onto HiTrap NHS-activated HP columns and using the ÄKTA purifier system (GE Healthcare) according to the instructions of the manufacturer.

### GST pulldown assay

GST-tagged *Drosophila* CYRI and CYRI$^{R163/163D}$ (pGEX 6P1; Addgene) were expressed in *ArcticExpress* (DE3) cells (Agilent Technologies) and purified via sonication in GST lysis buffer (2 mM MgCl$_2$, 2 mM DTT, 10% (vol/vol) glycerol in 1× PBS). GST-tagged proteins were then immobilized on Glutathione Sepharose 4B (Merck, Cytiva). S2 cells transfected with 3x-HA-tagged dRac1, dRac1$^{G12V}$, or dRac1$^{T17N}$ were harvested 72 h post-transfection. Cells were lysed in TLB buffer (50 mM Tris-HCL pH 7.4, 150 mM NaCl, 1.5 mM MgCl$_2$, 4 mM EDTA, 10% [vol/vol] glycerol, 1% [vol/vol] Triton X-100, 1 mM DTT) by vortexing. Protein lysates containing wild typic and mutant dRac1 were loaded with either GDP or GTPγS (Sigma-Aldrich) in a 10-min long reaction at 30°C. Once the reaction was stopped by adding 0.1 mM MgCl$_2$, GDP- or GTPγS-loaded dRac1 was incubated overnight at 4°C with GST-CYRI or GST-CYRI$^{R163/164D}$. Following the incubation, beads were washed and prepared for SDS PAGE and western blot analysis.

### SDS PAGE and Western blot analysis

Expression analysis of the cyriΔ2 and cyriΔ11 mutants was performed with lysates from isolated ovaries. 10 flies were dissected in 1× PBS and ovaries were carefully removed. Protein lysates were created by squashing the ovaries in lysis buffer (10 mM Tris pH 7.4, 100 mM NaCl, 2.5 mM MgCl2, 0.5% Triton X-100 + proteinase inhibitor) with a pestle. 4× SDS sample buffer was added to the supernatant and protein lysates were incubated for 10 min at 95°C.

Protein lysates were separated via SDS PAGE and analyzed by Western blot. The following antibodies were used: primary antibodies—anti-Tubulin (1:2,000; DSHB AA4.3), anti-GST (1: 5,000; Merck), anti-His-CYRI (1:250; purified), anti-GST-CYRI (1:250; purified), anti-GFP (1:1,000; ClonTech); secondary antibodies—goat anti-mouse IgG (H+L), HRP (1:5,000; Thermo Fisher Scientific), goat anti-rabbit IgG (H+L), HRP (1:5,000; Thermo Fisher Scientific), goat anti-guinea pig IgG (H+L), HRP (1:5,000; Thermo Fisher Scientific), donkey anti-goat IgG (H+L), and HRP (1:5,000; Thermo Fisher Scientific).

### Cell culture and cell transfection

*Drosophila* S2 cells were cultured in Schneider's medium as described in Stephan et al. (2008). Transfection of S2 cells was performed as described in Nagel et al. (2017). The following plasmids were used for transfection: pUAST attB CYRI, pUAST attB CYRI$^{R163/164D}$, pUAS attB GFP, pTWH dRac1-3x-HA, pTWH dRac1$^{G12V}$-3x-HA, and pTWH dRac1$^{T17N}$-3x-HA. All plasmids were created using the Gateway cloning system (Thermo Fisher Scientific).

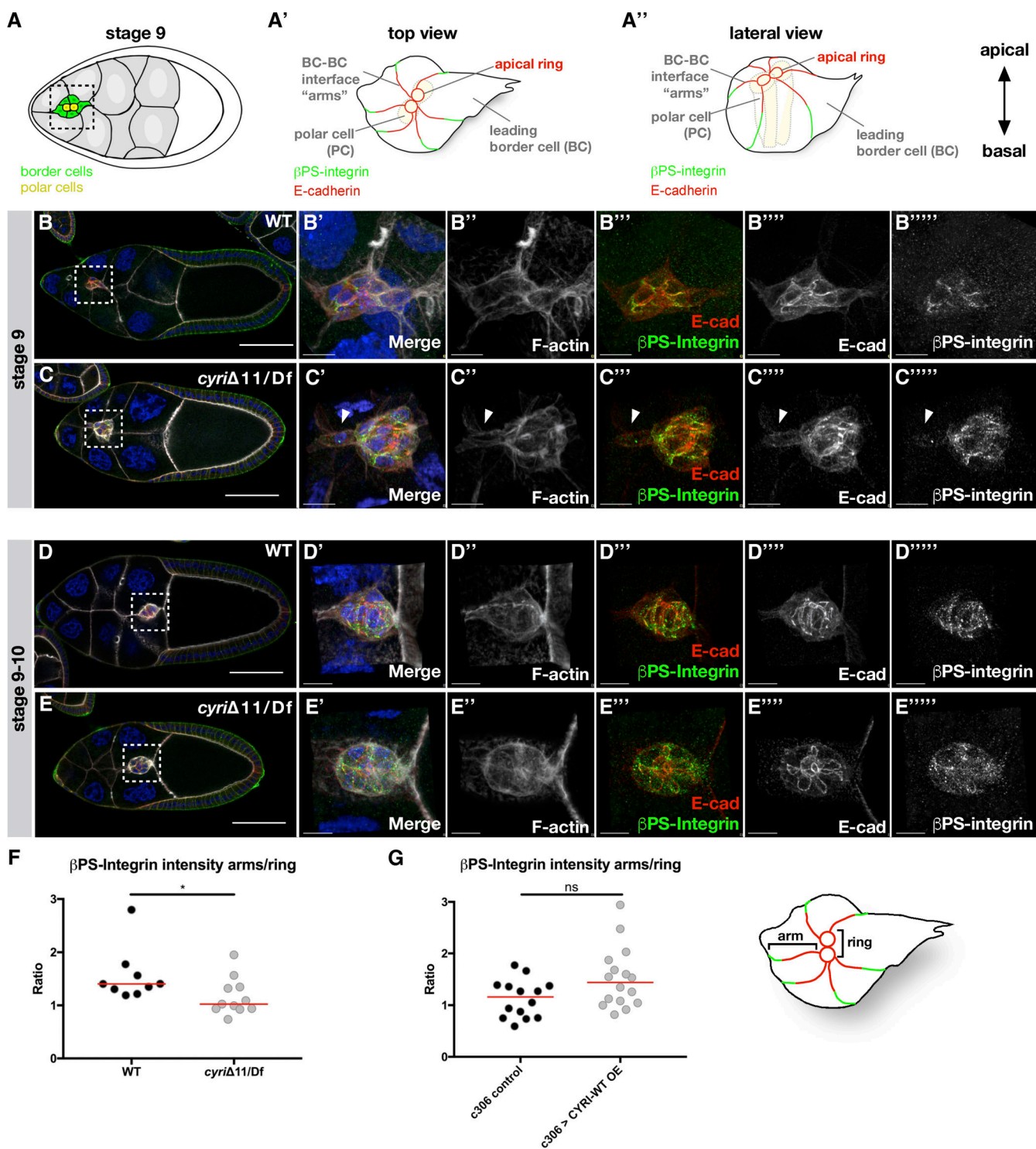

Figure 8. **Loss of CYRI affects β-integrin localization in border cell cluster. (A)** Schematic drawing of a stage 9 egg chamber. Polar cells are marked in yellow and border cells in green. **(A′ and A″)** Illustration of migrating border cell cluster that maintains the apico-basal polarity. The apical cap/ring structure is oriented approximately orthogonal to the direction of migration. Highest concentration of E-cadherin is found at the apical interface between border cells and polar cells (apical cap, ring-like structure) and at the contact side between adjacent border cells (BC-BC interface, "arms"). **(B–E)** Maximum intensity pro- jections of confocal slices of stage 9 egg chambers of the indicated genotypes (wild type: WT; *cyri* mutant: *cyri*Δ11/Df(ED4457)) with DNA (DAPI, blue), F-actin (Phalloidin, grey), anti-βPS-integrin (green), and anti-E-cadherin (red); anterior is to the left. **(B′–E″″″)** Detailed magnified 3D Imaris reconstructions of border cell clusters of boxed areas in B–E. The movement of the border cells proceeds from the left to the right. Scale bars represent 50 μm in B–E and 10 μm in B′–E″″″). **(F and G)** Quantification of βPS-integrin intensity of indicated genotypes. Statistical significance was determined using the Mann–Whitney test. **(F)** Trans- heterozygous *cyri* mutants (*cyri*Δ11/Df(ED4457): *n* = 11 and WT: *n* = 9, *P* value = 0.031. **(G)** Overexpression of wild type CYRI transgene under the control of the c306-Gal4 driver, c306 control: *n* = 14, c306 > CYRI-WT OE: *n* = 16, *P* value = 0.070. At least three independent experiments for each genotype were performed.

## Gel filtration chromatography

S2R+ cells were lysed in 25 mM Tris-HCl at pH 7.6, 100 mM NaCl, 2 mM MgCl$_2$, 0.5 mM EGTA, 5% glycerol, and a protease inhibitor mixture (Roche). Lysates were centrifuged 2 × 15 min at 16,000 $g$ to yield the cytoplasmic supernatant. The cytoplasmic supernatant was applied to a Superose 6 10/300 GL column (GE Healthcare). Collected fractions were precipitated with trichloroacetic acid and equal volumes of fractions were separated on standard SDS-PAGE. Proteins were analyzed by Western blots using an affinity-purified anti-WAVE antibody (guinea pig 1:2,000).

## CRISPR/Cas9-induced genomic GFP-tagging of proteins in S2 cells

GFP-tagging of endogenous CYRI in S2 cells stably expressing Cas9 nuclease (S2-Cas9) was performed as previously described (Böttcher et al., 2014). A DNA sequence encoding a GFP-tag was inserted at the 3′ end of the coding region of the *cyri* gene locus (targeting sequences: 5′-TTTGTTGGCCGCATAATAT-3′), leading to the expression of a C-terminally tagged protein. Briefly, S2-Cas9 cells were transfected with double-stranded linear DNA constructs encoding sgRNA and providing a template for homologous recombination (HR). PCR-based donor constructs were used as a template for homologous recombination (HR) after double-stranded breaks were introduced by the Cas9 nuclease. HR templates were amplified from plasmids containing GFP-tag sequences including a STOP codon and a resistance marker (pMH3, GFP-tag & Blasticidin resistance marker, a kind gift from A. Brehm).

This insert was amplified using primers containing 60 bp sequences homologous to regions directly up- and downstream of the original STOP codon (CYRI::GFP (endogen) antisense: 5′-GAAATTAACGTTAATGAACTCTCCGCCCACACCGGCCGCGCC CACCGATCCCCTCCAATAGAAGTTCCTATTCTCTAGAAAGTA TAGGAACTTCCATATG-3′) CYRI::GFP (endogen) sense 5′-ACG ACGAAGCATTTGAATGGGGAGAACACGCCGAAGAATATACAG CGTTTGTTGGCCGCAGGATCTTCCGGATGGCTCGAG-3′).

To promote double-strand break repair by HR, the protein amount of key enzymes involved in non-homologous end joining (NHEJ) and microhomology-mediated end joining (MMEJ) was lowered by transfecting S2-Cas9 cells with 1 µg/ml dsRNA targeting lig4 (NHEJ) and mus308 (MMEJ) transcripts. After 3 days, cells were transfected with HR and sgRNA templates using FuGENE HD transfection reagent (Promega). 4 days post transfection, cells were transferred to a medium containing 10 µg/ml Blasticidin (Gibco), respectively.

## Immunohistochemistry staining of pupal macrophages

Early pupae (0.5–4 h APF) were opened by gently pulling the epidermis apart with forceps, thereby releasing the hemolymph including macrophages into M3++ medium (Biomol). The cells were then placed on Concavalin A (Sigma-Aldrich) coated coverslips for 1 h at 25°C. Afterwards the macrophages were fixed using 4% PFA (Sigma-Aldrich) and washed with PBS. For antibody staining, the hemocytes were then incubated in 3% BSA (Roth) for 2 h at RT with the primary antibody (affinity-purified guinea pig antiserum, 1:1,000; Hirschhäuser et al., 2021) and for

45 min with the secondary antibody (anti-guinea pig Alexa488, Invitrogen), DAPI (Thermo Fischer Scientific), and Alexa568-coupled Phalloidin (Invitrogen). The coverslips were then mounted on object slides using Mowiol (Roth) and left to harden at 4°C for at least 1 h.

## Immunohistochemistry staining of egg chambers

Egg chambers were stained according to established protocol (McDonald and Montell, 2005). In detail, adult female flies were placed on fresh yeast for 24 h at 25°C or 29°C and then dissected in PBS. Ovarioles were isolated in Schneider medium and fixed in 4% PFA for 10 min. Egg chambers were washed and blocked in NP40 block (50 mM Tris pH7.4, 150 mM NaCl, 0.5% NP40, 5 mg/ml BSA) and incubated overnight with primary antibodies: anti-EYA (1:50; DSHB 10H6), anti-Ecad (1:50; DSHB DCAD2), and anti-β-Integrin (1:50; DSHB CF.6G11). Secondary antibodies used were goat anti-mouse Alexa Fluor488 and goat anti-rat Alexa Fluor568 (1:1,000; Invitrogen) and were supplemented with Alexa Fluor568 (1:200; Invitrogen) or Alexa Fluor647-conjugated Phalloidin (1:100; Invitrogen) and DAPI (1:100; Invitrogen). After 2 h of incubation at RT, Fluoromount-G was added and stored for at least 1 h at 4°C before mounting. Each experiment consists of 10 flies from at least three independent crossings.

## Bimolecular fluorescence complementation assay

Protein–protein interaction was investigated in vivo by bimolecular fluorescence complementation (Gohl et al., 2010). pUAST-Rac1-myc-NYFP, pUAST-CYRI-HA-CYFP, and pUAST-CYRI-$^{R163/164D}$-HA-CYFP were generated by introducing the corresponding cDNAs into the corresponding pUAST-BiFC vectors (Gohl et al., 2010) via Gateway-cloning (Invitrogen). YFP fragments were expressed under the control of the *en*-Gal4 driver in the posterior compartment of wing imaginal discs, whereas the anterior compartment served as a negative control. Wing imaginal discs were isolated from third instar larvae in ice-cold PBS and fixed in 4% PFA (Sigma-Aldrich) for 20 min and washed in PBT. After 30 min of incubation in blocking solution (3% BSA in PBT), incubation with primary antibodies was performed for 90 min, followed by three washing steps in PBT (20 min each). Secondary antibodies (anti-HA, 1:100; Invitrogen and anti-c-Myc, 1:10; DSHB 9E10) were added and incubated for 60 min in blocking solution, again followed by three washing steps in PBT (20 min each). Wing discs were finally mounted in mounting medium (Fluoromount-G; SouthernBiotech). Endogenous YFP and immunofluorescence were analyzed in confocal images acquired with the Leica TCS SP8 microscope.

## Image acquisition and microscopy

Confocal fluorescence images were acquired with the Leica TCS SP8 with an HC PL APO CS2 63×/1.4 and HC PL APO CS2 40×/1.3 oil objective. For cell spread measurements, macrophages were isolated from wandering third-instar larvae, pre-pupae, or pupae (0.5–4 h APF) and seeded on a glass surface coated with Concanavalin A (0.5 mg/ml; Sigma-Aldrich), and stained with phalloidin and DAPI. Wounding experiments of single and multiple (4–6) epithelial cells in the abdomen of pupae (18–20 h

APF) were acquired with the Zeiss CellObserver Z.1. Ablation experiments were done using a 355-nm pulsed UV provided by laser Rapp Optoelectronics. Wound closure was imaged for 60 min after wounding in single-cell ablations and for 75 min if multiple cells were ablated.

### Quantification of macrophage morphology, lamellipodia formation, wound closure of pupal epithelium, and egg chambers

The cell morphology of macrophages was analyzed with the polygonal selection tool or shape descriptors provided by FIJI (ImageJ, NIH). Wound closure and lamellipodial protrusions were analyzed with the area free-hand measurement tool of FIJI. In single-cell ablations, wound size was measured every 5 min and was normalized to the size of the wound after maximum expansion. In experiments with multiple ablated cells, the interval to measure wound size was increased to 10 min. The lamellipodial area was calculated by subtracting the area that was not covered by protrusion from the total wound area. Results were then normalized to the wound size at the same time point. Measurements for the lamellipodia formation were done every 5 min over the first 20 min after wounding. The wound healing was observed and measured for 60 min after wounding.

$$A_{lamellipodia\ at\ t_x} = \frac{\left(A_{total\ at\ t_x} - A_{without\ lamellipodia\ at\ t_x}\right)}{A_{total\ at\ t_x}} \times 100$$

For anti-mys intensity measurements in fixed egg chambers, lines were drawn along three membranes between outer border cells ("arms") and the membrane where polar and border cells connect (apical ring). The average intensity along the arms was divided by the intensity along the apical ring. The ratio was plotted and statistical significance was determined using the Mann–Whitney test.

### Statistics

Experiments were repeated in at least three independent experiments. Statistical analysis was performed with GraphPad Prism software Version 5 and 8. The normal distribution of data sets was tested via the Shapiro–Wilk test. In wound healing and lamellipodia size, normal distribution was confirmed through graphical analysis of QQ-plots. Analysis of hemocyte size and intensity measurements was performed using the t-test for normally distributed data sets of two unpaired groups or using the non-parametric Mann–Whitney U test for data sets that were not normally distributed. Border cell migration defects were analyzed using Fisher's exact or Chi$^2$ tests, depending on the number of observed phenotypes. We usually used one-way analysis of variance (ANOVA) for comparing means in a situation where there are more than two groups, except wound healing experiments were analyzed using two-way ANOVA.

### Online supplemental material

The associated supplemental files contain four figures and eight videos. Fig. S1 is related to Fig. 1, showing that *Drosophila* CG32066 is the ortholog of human CYRI. Fig. S2 is related to Fig. 5, showing that MBT-GFP sensor specifically binds GTP-loaded *Drosophila* Rac1 and Rac2. Fig. S3 is related to Fig. 5, showing that MBT-GFP sensor specifically binds GTP-loaded Drosophila Rac1 and Rac2. Fig. S4 is related to Figs. 6 and 7, showing that overexpression of activated WAVE results in border cell cohesion defects. Video 1 is related to Fig. 1, showing a similar predicted 3D structure of the *Drosophila* CYRI protein and its human orthologues. Video 2 is related Fig. 2, showing that endogenous CYRI localizes at leading pseudopods. Video 3 is related Fig. 4, showing that the WRC-Arp2/3 pathway is required for wound closure. Video 4 is related Fig. 5, showing that loss of CYRI accelerates epidermal wound closure. Video 5 is related Fig. 5, showing *cyri* mutant wound closure defects, rescued by re-expression of wild type CYRI but not CYRI$^{R163/164D}$ variant. Video 6 is related Fig. 5, showing increased MBT-GFP intensity in *cyri*-depleted epidermal cells upon wounding. Video 7 is related Fig. 5, showing increased Abi-GFP intensity in *cyri*-depleted epidermal cells upon wounding. Video 8 is related Fig. 8, showing a 3D Imaris reconstruction movie of wild type and *cyri* mutant border cell clusters.

### Data availability

The data are available from the corresponding author upon reasonable request.

## Acknowledgments

We acknowledge the Bloomington Drosophila Stock Center and Vienna Drosophila Resource Center for fly stocks. We thank Alexander Hirschhäuser for performing BIFC experiments and structural modeling. We thank Katja Rust for thoughtful discussions and critical reading of the manuscript and Darius Molitor for general support.

The work was supported by grants to S. Bogdan from the Deutsche Forschungsgemeinschaft (DFG). Open Access funding provided by Philipps-Universität Marburg.

Author contributions: M. Rotte: Formal analysis, Investigation, Methodology, Resources, Validation, Visualization, Writing—review & editing, M.Y. Hohne: Formal analysis, Investigation, Validation, Visualization, Writing—review & editing, D. Klug: Data curation, Formal analysis, Investigation, Validation, Visualization, Writing—review & editing, K. Ramlow: Investigation, Methodology, Validation, C. Zedler: Formal analysis, Investigation, Visualization, F. Lehne: Formal analysis, Investigation, Writing—review & editing, M. Schneider: Data curation, Formal analysis, Investigation, Validation, Visualization, Writing—review & editing, M.C. Bischoff: Investigation, Validation, S. Bogdan: Conceptualization, Data curation, Funding acquisition, Investigation, Methodology, Project administration, Supervision, Validation, Visualization, Writing—original draft, Writing—review & editing.

Disclosures: The authors declare no competing interests exist.

Submitted: 30 October 2023

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

**Supplemental material**

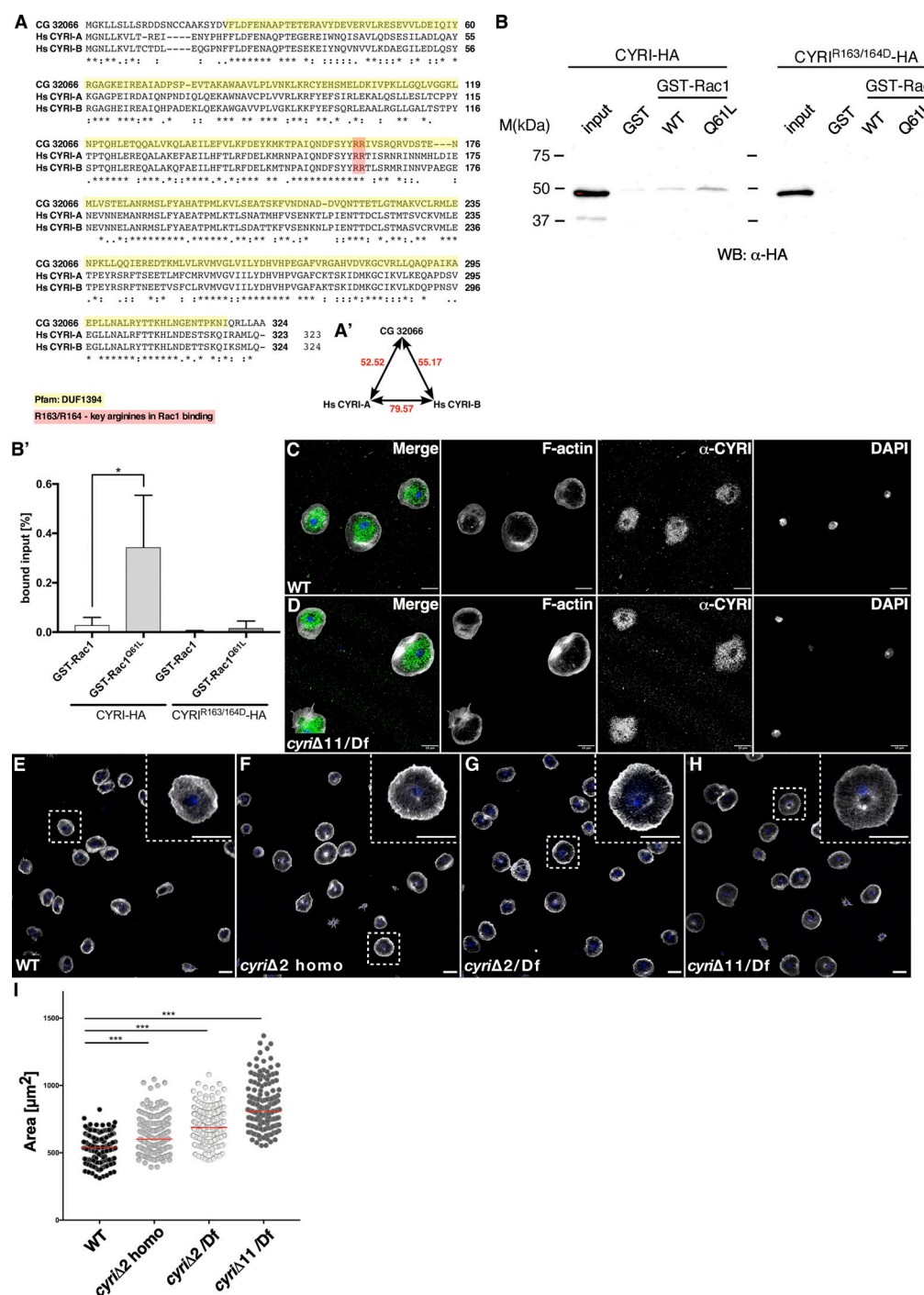

Figure S1.  **Drosophila CG32066 is the ortholog of human CYRI. (A)** Sequence alignment of human CYRI-A, CYRI-B and *Drosophila* CG32066. Conserved amino acid residues are marked by asterisks. Colons indicate conserved positions containing residues with strongly similar properties. Gaps are indicated by dashes. The conserved two arginine residues, R163 and R164 are highlighted in red. The highest conservation is found within the DUF1394 domain (highlighted in yellow). **(A′)** The *Drosophila* CYRI protein shares 52.52% identity with the human (Hs) CYRI-B protein and 55.17% identity with the human (Hs) CYRI-A protein. The human proteins share 79.57% identity. **(B)** Western blot from pull-down of GST control, GST-Rac1WT, or GST-Rac1Q61L beads, with cell lysate expressing either HA-tagged wild type CYRI or HA-tagged mutant CYRI-R163/164D variant. **(B′)** Quantification of (B) from three independent experiments. Signals were normalized to loaded input (1% of the starting lysate material). Mean ± SD. Statistical analysis using one-way ANOVA with Tukey's multiple comparisons. * P < 0.05. **(C and D)** Confocal images of macrophages isolated from (C) wild type and (D) trans-heterozygous *cyri* mutant pupae (*cyri*Δ11/Df[ED4457]), stained with phalloidin-Alexa488 (grey), DAPI (blue), and an anti-CYRI antibody (green). Note: no differences in immunofluorescent anti-CYRI intensity were detected. **(E–H)** Confocal images of (E) wild type and (F) homozygous *cyri*Δ2 mutant (G) transheterozygous *cyri*Δ2/Df(ED4457) mutant and (H) transheterozygous *cyri*Δ11/Df(ED4457) mutant macrophages were co-stained with phalloidin (grey) and DAPI (blue). High magnification of boxed areas displays single cells. Scale bars represent 10 µm. **(I)** Quantification of spread cell area, *n* = 167 for all genotypes. To evaluate statistical significance, one-way-ANOVA (Kruskal–Wallis test with Dunn´s correction) was used. P value: <0.001 (***). The red bar represents the median. Three independent experiments for each genotype were performed. Source data are available for this figure: SourceData FS1.

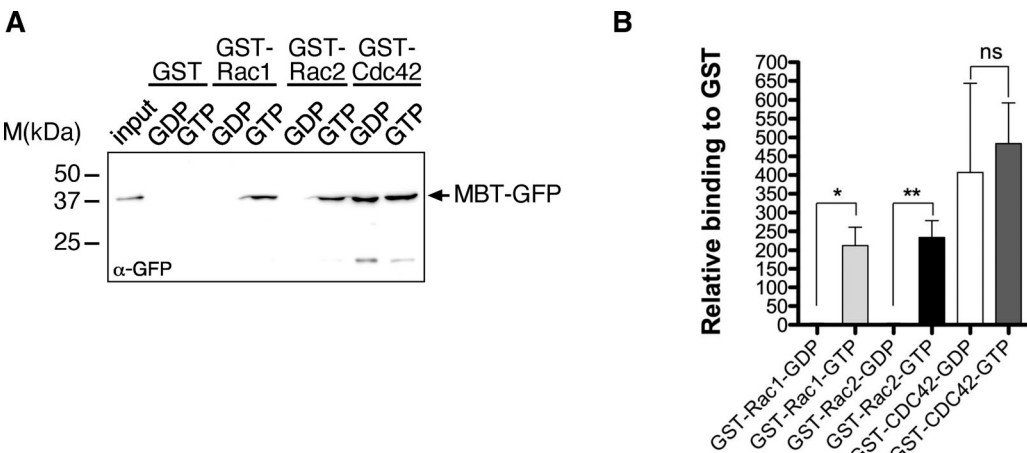

Figure S2.  **A new GFP-based sensor for active Rac1 and Rac2. (A)** Pull-down experiments with recombinant GST, GST-Rac1, -Rac2, and -Cdc42 proteins. GSH-sepharose-bound GST-Cdc42 was preloaded with GDP or GTP γS and incubated with S2 cell lysate transfected with MBT-GFP. Bead-bound complexes were probed for binding of MBT-GFP protein by an anti-GFP antibody. **(B)** Quantification of (A) from three independent experiments. Signals were normalized to GST. Mean ± SD. To evaluate statistical significance, the Mann-Whitney test was used and following P values (two-tailed) were obtained: P value: ** <0.001, * <0.05 and ns > 0.05. Source data are available for this figure: SourceData FS2.

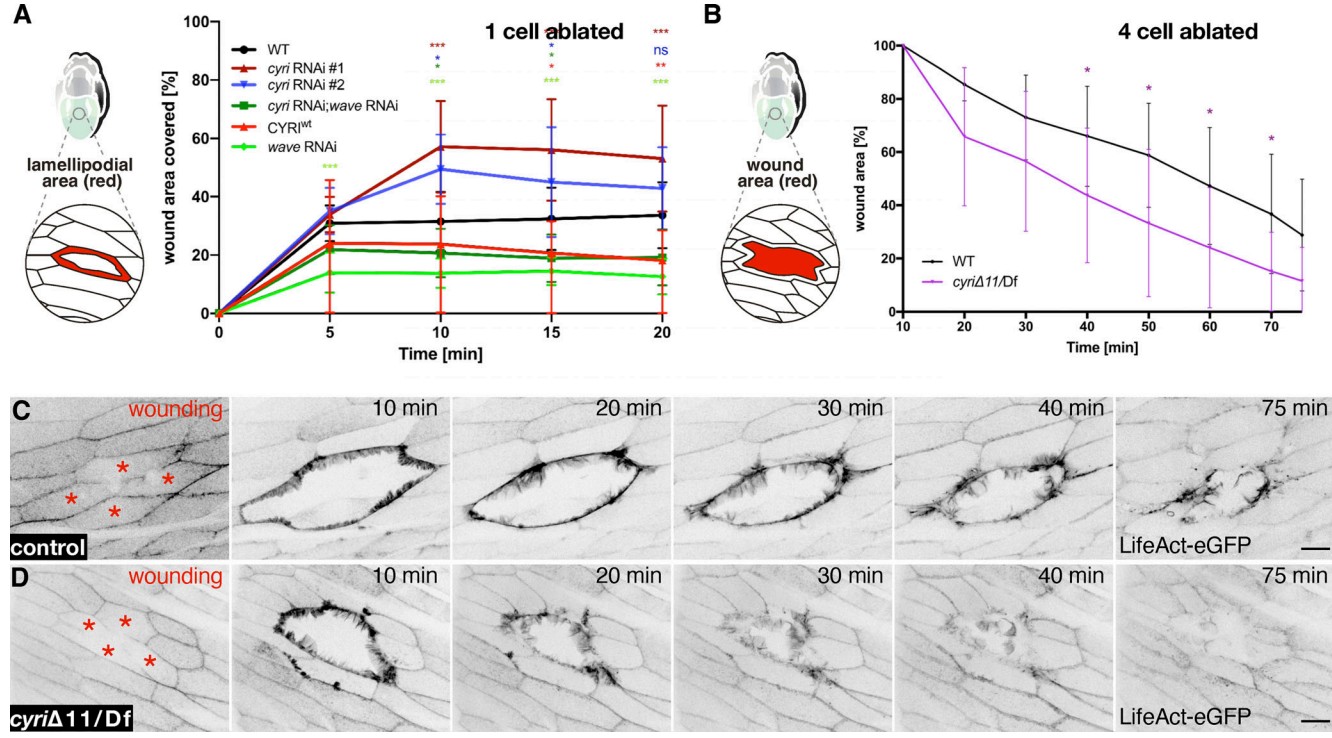

Figure S3.  **Suppression of cyri function promotes epidermal wound closure. (A)** Quantification of increased lamellipodial area upon expression of two different *cyri* RNAi transgenes (#1 and #2, n = 5), *wave* RNAi (n = 16), *cyri; wave* double RNAi (n = 12), CYRI^WT (n = 15) under the epidermis-specific A58-Gal4 driver. Note that wild type control (n = 13) and *wave* RNAi are the same as in Fig. 4, F and G and CYRI^WT is the same as Fig. 5 E', as the data belong to the same dataset. Lamellipodia size was measured every 5 min and normalized to the initial size of the unwounded cell. The two-way ANOVA analysis with Dunnett correction was used and P values were obtained: P value: 0.12 (ns), 0.033 (*), 0.002 (**), <0.001 (***). **(B–D)** Multiple-cell ablation experiments in the abdominal epidermis of wild type and trans-heterozygous *cyri* mutant pupae (*cyri*Δ11/Df[ED4457]), marked by the expression of Lifeact-EGFP under the control of the A58-Gal4 driver. The data shown represent the wound area at the cell surface to illustrate the effects of rapid ingrowth of lamellipodia on wound size (B) Quantification of wound closure in wild type control (black) and trans-heterozygous *cyri* mutant pupae (purple). Wound size was measured every 10 min and normalized to the initial size of the unwounded cell, (n = 5). To evaluate statistical significance, the two-way-ANOVA Bonferroni post test, (P < 0.05) was used. Error bars represent SD. **(C and D)** Frames of spinning disc microscopy videos of 18 h APF old epidermis expressing a Lifeact-EGFP transgene under the control of the A58-Gal4 driver. The genotypes are indicated. Red asterisks mark ablated epidermal cells. Images were taken at indicated timepoints. Scale bar represents 20 μm.

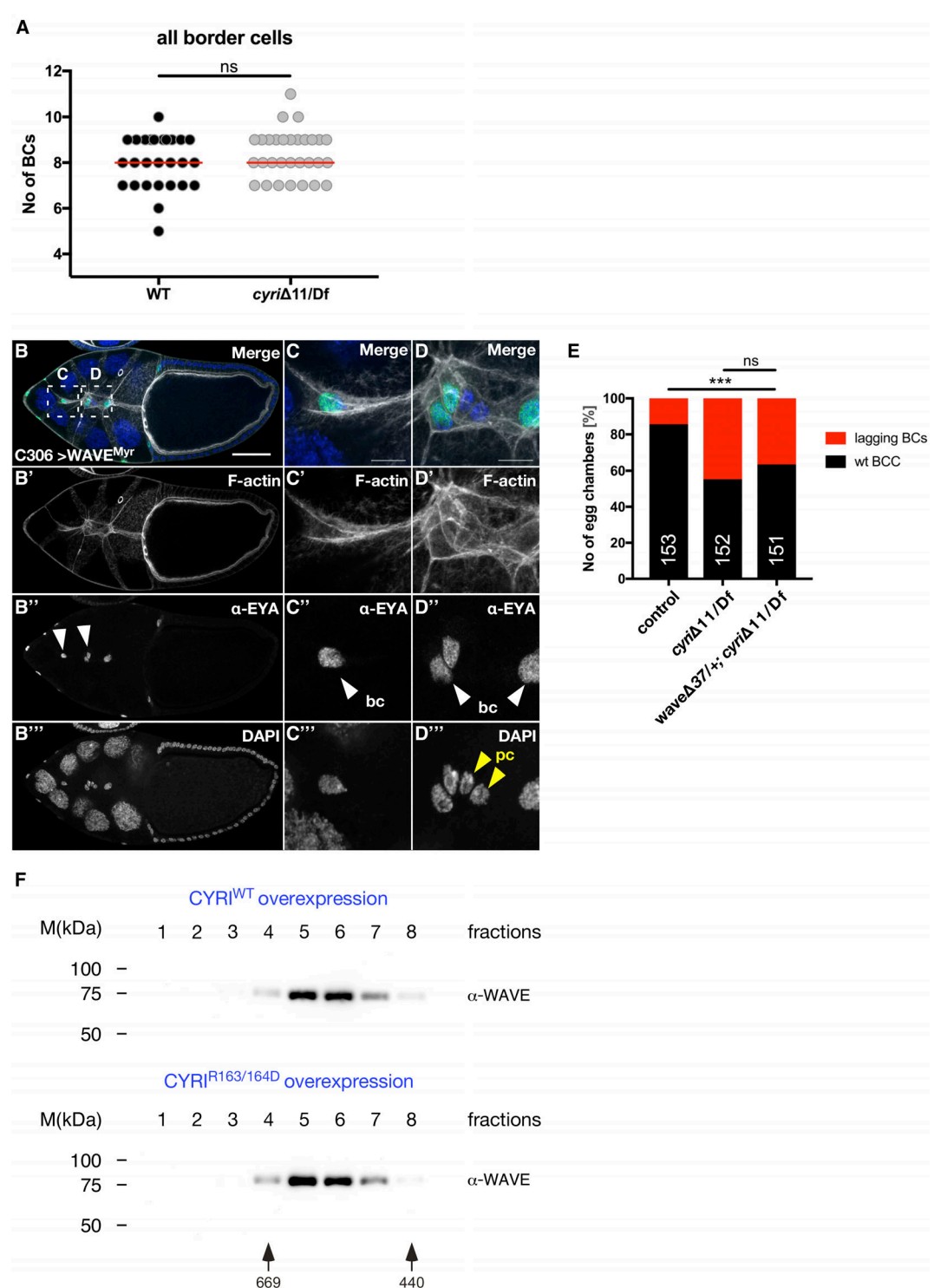

Figure S4. **Overexpression of activated WAVE results in border cell cohesion defects. (A)** Quantification of border cell numbers in wild type and trans-heterozygous *cyri* mutant egg chambers. Statistical significance was tested using the Mann–Whitney test, P value = 0.343. WT: *n* = 27, *cyri*Δ11/Df: *n* = 30. Three independent experiments for each genotype were performed. **(B–D)** Maximum intensity projections of five confocal slices of stage 9 egg chambers over-expressing a membrane-tethered WAVE construct (WAVE^Myr) under the control of the C306 driver, stained for DNA (DAPI, blue), F-actin (phalloidin, grey), and anti-EYA (green); anterior is to the left. **(B)** Maximum intensity projection of a stage 10 egg chamber overexpressing WAVE^Myr. Scale bars represent 50 μm. White arrowheads mark border cells (bc). **(C and D)** Detailed views of boxed areas in B show an abnormally elongated border cell cluster and border cohesion defects. White arrowheads mark border cells (bc) whereas polar cells (pc) are marked by yellow arrowheads. Scale bar represents 10 μm. **(E)** Removal of one copy of *wave* in *cyri* mutant background reduced the lagging border phenotype, although the difference was not significant. N numbers are indicated, P value: ns = 0.1607, ***<0.001. **(F)** Gel filtration profiles of endogenous WAVE complexes from S2R+ cells overexpressing wild type CYRI (CYRI^WT) and Rac-binding deficient variant (CYRI^R163/164D) constructs. Complexes co-fractionated with high molecular weight complexes at 500–700 kDa sizes. The elution profile of proteins of known molecular mass is indicated at the bottom. Source data are available for this figure: SourceData FS4.

Video 1. **Conserved 3D structure of CYRI proteins using the UCSF Chimera software (**Pettersen et al., 2021**).** An overlay of the crystal structure of the dimeric mouse CYRI-BΔN (26–324 aa) lacking the first 25 amino acids (PDB:7AJL; light blue; Yelland et al., 2021) with *Drosophila* CYRI (Q7K1H0; green) and human CYRI-B (Q9NUQ9; magenta) based on AlphaFold2 protein structure predictions is shown. Note the model contains the two highly conserved arginines at positions 163 and 164 (marked in blue) in the fly protein corresponding to R161 and R162 in human CYRI-B. 10 frames per second (fps).

Video 2. **Leica TCS SP8 microscopy time-lapse video of *Drosophila* S2 cells in which the endogenous cyri locus was tagged by a GFP fusion.** Endogenous GFP-tagged CYRI protein, only expressed at a very low level, localizes at leading pseudopods. Images were taken every 5 s for 2 min. 10 frames per seconds (fps).

Video 3. **Spinning-disk microscopy videos of the abdominal epidermis 18 h APF old pupae with indicated genotypes (WT, wild type and after knockdown of *arp*2, *arp*3 and *wave* by RNAi) specifically expressing a Lifeact-eGFP transgene under the control of the *A58*-Gal4 driver.** Images were taken every 30 s for 60 min, ablation starts at *t* = 0 min. Scale bar represents 50 µm. 15 frames per seconds (fps).

Video 4. **Spinning-disk microscopy videos of the abdominal epidermis 18 h APF old pupae with indicated genotypes (overexpressing a wild type CYRI, mutant CYRI^R163/164D transgene, homozygous *cyri*Δ2 mutant and transheterozygous *cyri*Δ11/Df(ED4457) mutant) specifically expressing a Lifeact-EGFP transgene under the control of the *A58*-Gal4 driver.** Images were taken every 30 s for 60 min, ablation starts at *t* = 0 min. Scale bar represents 50 µm. 15 frames per seconds (fps).

Video 5. **Spinning-disk microscopy videos of the abdominal epidermis 18 h APF old pupae with indicated genotypes right *cyri*Δ11/Df(ED4457 mutant; middle: *cyri*Δ11/Df(ED4457 mutant re-expressing a wild type CYRI; left: *cyri*Δ11/Df(ED4457 mutant re-expressing the CYRI^R163/164D variant.** Images were taken every 30 s for 60 min, ablation starts at *t* = 0 min. Scale bar represents 50 µm. 15 frames per seconds (fps).

Video 6. **Spinning-disk microscopy videos of the abdominal epidermis 18 h APF old pupae expressing the MBT-GFP transgene under the control of the *da*-Gal4 driver.** Images were taken every 30 s for 30 min, ablation starts at *t* = 0 min. Scale bar represents 50 µm. Left (*da*-Gal4) control, right *cyri* RNAi transgene. 10 frames per second (fps).

Video 7. **Spinning-disk microscopy videos of the abdominal epidermis 18 h APF old pupae expressing the Abi-GFP transgene under the control of the *da*-Gal4 driver.** Images were taken every 30 s for 30 min, ablation starts at *t* = 0 min. Scale bar represents 50 µm. Left (*da*-Gal4) control, right *cyri* RNAi transgene driven by *da*-Gal4. 15 frames per second (fps).

Video 8. **3D Imaris reconstruction movies of wild type and *cyri* mutant border cell clusters stained for DNA (DAPI; blue), F-actin (phalloidin; grey), anti-βPS-integrin (green) and anti-E-cadherin (red); anterior is to the left.** Scale bar 10 µm. 15 frames per second (fps).

