## [Peer Review File · The Journal of Cell Biology]

CYRI controls epidermal wound closure and cohesion of invasive border cell cluster in *Drosophila*

Marvin Rötte, Mila Höhne, Dennis Klug, Kirsten Ramlow, Caroline Zedler, Franziska Lehne, Meike Schneider, Maik Bischoff, and Sven Bogdan

Corresponding Author(s): Sven Bogdan, University of Marburg

Review Timeline:

Submission Date:	2023-10-30
Editorial Decision:	2023-12-12
Revision Received:	2024-07-19
Editorial Decision:	2024-09-05
Revision Received:	2024-09-19

Monitoring Editor: William Bement

Scientific Editor: Tim Fessenden

Transaction Report:

DOI: <https://doi.org/10.1083/jcb.202310153>

December 12, 2023

Re: JCB manuscript #202310153

Prof. Sven Bogdan
University of Marburg
Institute of Physiology and Pathophysiology
Marburg 35037
Germany

Dear Prof. Bogdan,

Thank you for submitting your manuscript entitled "CYRI controls epidermal wound closure and cohesion of invasive border cell cluster in *Drosophila*". The manuscript has been evaluated by expert reviewers, whose reports are appended below. Unfortunately, after an assessment of the reviewer feedback, our editorial decision is against publication in JCB.

You will see that reviewer enthusiasm for these otherwise interesting findings was reduced by two issues: the observation that these findings extend known functions of CYRI in flies (Reviewer 1), and the lack of mechanistic detail needed for a thorough understanding of how CYRI contributes to wound healing and collective migration in the two physiological settings examined (Reviewers 2 and 3). Reviewer 2 also sought validation, controls, and quantification of multiple observations.

Although your manuscript is intriguing, the points raised by the reviewers are more substantial than can be addressed in a typical revision period. If you wish to expedite publication of the current data, it may be best to pursue publication at another journal.

Given our interest in this topic, we would be open to resubmission to JCB of a significantly revised and extended manuscript that fully addresses the reviewers' concerns and is subject to further peer-review. A satisfactory revision would include all changes requested by reviewers 2 and 3. We expect that the resolution of their concerns would offer sufficient novelty into CYRI functions to overlook the concern of Reviewer 1. In addition, we agree with Reviewers 1 and 3 that the introduction should be improved in a revised manuscript. If you would like to resubmit this work to JCB, please contact the journal office to discuss an appeal or you may submit an appeal directly through our manuscript submission system. An appeal document must include a plan for revision which we may discuss with reviewers. Please note that priority and novelty would be reassessed at resubmission.

Regardless of how you choose to proceed, we hope that the comments below will prove constructive as your work progresses. We would be happy to discuss the reviewer comments further once you've had a chance to consider the points raised in this letter. You can contact the journal office with any questions at cellbio@rockefeller.edu.

Thank you for thinking of JCB as an appropriate place to publish your work.

Sincerely,

William Bement
Monitoring Editor
Journal of Cell Biology

Tim Fessenden
Scientific Editor
Journal of Cell Biology

Reviewer #1 (Comments to the Authors (Required)):

The manuscript by Bogdan and colleagues describes the contribution of CYRI, a Rac1-interacting protein, to epithelial cell behaviors in vivo. CYRI was identified as an inhibitor of the WAVE regulatory complex a few years ago, and its function is mostly understood in cell culture or in conditional mutants that affect immune cells of mice, leaving open questions of its role in epithelia. The authors begin by identifying the *Drosophila* homolog, demonstrating its physical interaction with Rac1, and analyzing its function with respect to the actin cytoskeleton in insect cell culture, where it behaved as expected in inhibiting lamellipodial protrusions and promoting spiky filopodial protrusions. Next, they generated two fly mutants with CRISPR, both

viable. However, when wounded, the stronger of these mutants, CYRI[11]/Df, closed wounds faster than wild-type controls, whereas overexpression of CYRI slowed wound closure, similar to loss of WRC proteins. Noticing that these mutants had reduced offspring, the authors analyzed defects in oogenesis, determining that border cell migration was disrupted with individual cells lagging and not reaching the nurse cell-oocyte boundary, resulting in a reduced micropyle. This phenotype was stronger than that caused by the overexpression of WAVE, suggesting an additional function for CYRI in border cell migration.

My overall assessment is that this is a solid paper that covers a lot of ground, but it is suited for a more specialist journal than JCB. There is a lot of rigorous work presented, geared generally to addressing the question, what does CYRI do? Many of the experiments and results confirm previous findings (acknowledged by the authors), which is reassuring, including the data in Figs. 1-3. The main contribution of the manuscript is in Fig. 5, where the CYRI wound healing data is presented, and in Fig. 4 containing the WRC data, a companion to Fig. 5. The data on border cell migration in Figs. 6 and 7, although interesting and generally confirming the function of CYRI in collective epithelial cell migrations, points in a new direction - integrin regulation by CYRI - which is not analyzed in any depth. I wish the authors had instead focused on either wound healing or border cell migration to understand one phenomenon more fully.

The paper has a rocky start with way too much molecular detail in the first paragraph, and I strongly suggest that the authors rewrite the introduction, focusing on the bigger picture of actin protrusions, when they are needed in individual cells and in epithelia, and feed-forward and feedback loops that control them. If the molecular details that are currently presented in the first paragraph of the introduction are necessary, they need an accompanying figure.

Reviewer #2 (Comments to the Authors (Required)):

The manuscript by Rötte and colleagues describes the characterisation of CYFIP in *Drosophila* epidermal and epithelial tissues to better understand its function in cell dynamics *in vivo*.

This study builds on recent work from the Machesky lab, which has reported that CYFIP-related Rac1 interactor (CYRI) proteins compete with the WAVE regulatory complex (WRC) for interaction with Rac1 in a feedback loop, regulating lamellipodia dynamics. However, the physiological role of CYRI proteins in normal tissues had not been established. This study adds to the knowledge about CYRI by addressing this issue in a well-studied and tractable model system.

Although not essential for viability, this study finds that in CYRI controls lamellipodia spread in isolated macrophages, wound healing in epidermal cells and collective cell migration in migratory epithelial cells of the ovary. The genetic data and molecular analysis presented is consistent with *Drosophila* CYRI acting as a regulator of WRC, although some uncertainty remains about its precise mode of action.

Overall, this manuscript presents new findings on the physiological role of a CYRI protein, which has previously been described as a Rac1-binding regulator of lamellipodia and macropinocytic events. The manuscript is generally well-written. Most of the conclusions are supported by the data presented. However, some aspects appear more preliminary due to lack of quantification and incomplete statistical analyses.

The following weaknesses should be addressed:

1. Sample sizes are not always stated, statistical analyses are missing or are not appropriate, and the variation for each genotype is not clearly displayed in figures or reported in the text. The authors should clarify how many different times independent experiments were performed, and how many flies were analysed per experiment. At least three independent experiments should be carried out for each genotype.
2. When comparing more than two conditions, the authors should use either non-parametric, Kruskal-Wallis test with Dunn's Multiple Comparison test or one-way ANOVA with Tukey Multiple Comparison test to reveal statistically significant differences between datasets throughout, depending on whether the data are distributed normally. Mann-Whitney and T tests are inappropriate when there are more than two independent samples. It is critical to measure the variation within and between experiments to identify significant differences between different genotypes.
3. While the data in Figure 1 suggest CG32055/*Drosophila* CYRI can interact with activated Rac1, the Western blots and immunofluorescent images in Figure 1 need to be quantified, along with biological repeats, and the data subjected to appropriate statistical analysis. Further, the data presented is based entirely on overexpression, both in cells (Fig.1D-E) and in imaginal discs (Fig.1F-G). If possible, the authors should attempt to demonstrate an interaction between endogenous proteins takes place, to increase confidence that CYRI is a Rac1 binder *in vivo*.
4. The experiments examining the subcellular localisation and effect of CYRI on lamellipodial protrusions are somewhat preliminary. The anti-CYRI staining in Fig 2A is rather weak. Although the antibody appears to specifically detect the untagged protein on Western blots (Figures 2B and 3B) and a proportion of cells ectopically expressing CYRI (Fig.2F-G), it would be useful to show that the immunofluorescent signals in Fig.2A are lost after RNAi-knockdown. Does CYRI colocalise with F-actin in Fig.2A, or with Scar/WAVE?
5. Noticeably, the distribution of the CYRI antibody stain is different from the distribution of GFP-CYRI in Fig.2D', where there appears to accumulation in the nucleus and some signal at the cell periphery. This might suggest that the distribution of the

GFP-CYRI protein has been affected by the GFP tag - failure to detect the protein on a blot at least suggests a change in accessibility of the anti-CYRI epitope. Although I appreciate the effort to make the CRISPR line, I'm not sure what it adds unless the distribution can be shown to faithfully recapitulate the endogenous untagged protein.

6. The authors show that ectopic expression of the CYRI protein changes the cell morphology in a proportion of S2 cells (Fig2F-G). Some cells appear unaffected, but this is probably related to CYRI expression level/transfection efficiency. It would be useful to quantify this relationship. Furthermore, the quantification in Fig2. is impossible to interpret without knowing the number of biological repeats and distribution of the data (see point 1).

7. To strengthen the implied mechanism by which ectopic CYRI proteins are acting, it would be helpful to examine whether ectopic expression of CYRI mutant or CYRI RNAi knockdown results in enlarged or broad lamellipodia as it does in human cells. This would also strengthen the findings in Fig.3. Is the recruitment of Scar/WAVE affected as it is in human cells? Furthermore, is ectopic CYRI wt capable of suppressing any Rac1-hyperactivation phenotypes, including increased cell circularity?

8. The conclusions in Figure 3 seem well supported, including genetic evidence from transheterozygous and hemizygous mutant animals that loss of CYRI in flies is not essential for viability. Although only found in one genotype (*cyri[d11]/Df*), the cell spreading phenotype looks convincing and is reminiscent of the *cryi-b* knockdown phenotype in mammalian cells. Again, staining of these cells with Scar/WAVE proteins to examine its recruitment to lamellipodia would help to confirm the underlying mechanism.

9. In the abstract, the authors state that *cyri* controls both lamellipodia spread and protrusion dynamics. However, as far as I can see there is no analysis of protrusion dynamics. It would be useful to show effects on dynamics in any one of the experimental systems.

10. Experiments analysing the effect of CYRI and Rac-WRC-Arp2/3 pathway during wound closure are informative and support the proposed model. However, it looks like the control in Fig.4F-G is the same as the one in Fig.5E-F. I can understand that the data have been presented in this way for clarity. However, this issue needs to be clarified in the figure legends and the statistical tests should be run across all genotypes using the appropriate multiple comparison test to take account of variance (see point 2).

11. Switching to a model of collective migration, the authors show that *cyri[d11]/Df* flies display a border cell migration defect, which may explain an observed reduction in fecundity. It is impossible to tell from these experiments whether the requirement for *cyri* is in just the migratory outer border cells or whether *cyri* also plays a role in non-migratory polar cells. Furthermore, given the reliance on just *cyri[d11]/Df* for phenotypic information in this context, it would be reassuring to phenocopy the effect with *cyri* RNAi or look for complementation with *cyri* wt overexpression specifically in the outer border cells.

12. It is interesting that membrane tethered activated WAVE had a weaker effect than *cyri* loss of function, leading the authors to test the involvement of *cyri* in regulating cluster cohesion via integrin-mediated adhesion. Le et al (2021), previously reported that CYRI proteins affect integrin trafficking, so perhaps differences in integrin distribution is not unexpected. The parallels with this study should be mentioned in the text. The authors should test whether this role is Rac dependent, by testing for a lack of complementation with their UAS-*cyri* mutant, compared to the wild type transgene.

Minor issues

Line 192: some general information about the gene, e.g. the cytological location, doesn't seem particularly relevant and can be removed.

Line 357-361: this is a long sentence interrupted with a full-stop.

Line 375: The authors should clarify the following statement, which could seem counterintuitive to the reader "As in cultured mammalian cells, *Drosophila* CYRI broadly localized in the cytoplasm. Thus, CYRI behaves like a 'local inhibitor' of WRC at lamellipodial tips by restricting Rac1's activity at the cell membrane."

Reviewer #3 (Comments to the Authors (Required)):

Summary

This manuscript from Rotte et al characterizes the *Drosophila* ortholog of the mammalian proteins, CYRI-A/B, which can regulate actin dynamics through altering WAVE and Rac activity. This gene has not been previously characterized in flies, so the paper makes a good contribution and provides foundation for further study. Rotte et al examine the expression of CYRI in S2 cells in culture, and generate tools to characterize it further, specifically they made an antibody, generated two mutant alleles via CRISPR, and two inducible overexpression strains, one wild type and one mutant. CYRI interacted with Rac1 in pull-down assays and physically in wing disc. *Cyri* overexpression collapsed lamellopodia in cell culture, which phenocopies loss of wave function, and this required Rac1-binding-residues in CYRI. The overexpression phenotypes resemble those for loss of wave components in an epithelial wound healing assay. Interestingly, *Cyri* gain or loss of function could control the speed at which the

wound closes. Additional cyri mutant defects are described for ovarian border cell migration.

Generally the manuscript is very clear and well written, and the data are presented well. However, some of the data is circumstantial and not definitive, additional experiments are needed to pin down what is happening with the cyri mutants and to connect CYRI and a functional mechanism. I provide more details below.

Major comments

The authors generate two loss of function mutant alleles, and determine that delta2 is a hypomorph and delta11 is a null. They test these in various combinations and in trans to deficiencies. However, they do not rescue the mutant defects by putting the gene back, which is needed to confirm the phenotypes they see are due to the cyri mutations. This is especially important since they acknowledge there seems to be an off-target hit that makes the delta 11 allele homozygous lethal. I understand that overexpression has defects on its own, but it could be expressed at low levels.

In wound closure, gain of CYRI can inhibit or promote cell dynamics in ways that are similar to knockdown of WAVE or Arp2/3. But this correlative data does not necessarily mean they work in the same pathway. The argument would be more convincing if they did genetic interaction tests here, or tried to overcome knockdown of wave by genetic reduction of cyri, for example. These or similar experiments are needed to substantiate the claim in the abstract that CYRI is a "potent WRC regulator."

The authors use cyri mutant egg chambers to study border cell migration, which means that both border cells and the nurse cell substrates would be affected. To interpret this phenotype clearly, one needs to know in which cell type(s) the gene is acting. The authors may be able to use their antibody to see where in the egg chamber CYRI is expressed, or they could do cell autonomous knockdown or rescue experiments in some cells using the lines they have. It would be interesting to see if CYRI overexpression has an effect in border cells and if the R163/164D mutant could rescue the loss of function. In addition, it would again be helpful to look for genetic interactions between mutants for CYRI and the predicted interacting proteins, eg, wave or Rac to implicate them in functional interactions.

I found the changes to integrin staining subtle and hard to see. How was this measured for figure 7F? It would be better to normalize the levels against something not likely to change (eg, membrane GFP), or even more convincing to see compensatory changes in integrin when cyri was overexpressed in border cells.

Minor comments

The introduction has a lot of jargon and is a bit challenging to follow without knowing a fair amount about WAVE regulatory complex and Rac already.

The authors made an antibody, but it doesn't detect the GFP-fusion protein, which is odd. Maybe they can explain this?

Can the authors use the expression analysis tools (antibody or GFP) to look where CYRI is expressed in pupal epithelia? Does it localize to the cortex of cells under normal conditions? Does it localize to the actinomyosin ring? These data may also help make a clearer case for CYRI function.

Line 242/3 seems like a word is missing? "...compensate progressively (for) initial wound closure..."

Line 278 should be 6D not 5D

Figure panel 2H should have n values.

How does the R163/164D mutant CYRI affect border cells?

The text states (line 286/7) "mutant border cell clusters that reached nurse cell-oocyte boundary often contained less migratory border cells (Figure 6G3). However, this was not quantified. The activated wave phenotype in border cells appears to be much stronger in terms of fewer cells overall. These border cells should be quantified in both cases. It is an important point since in mutants that have additional border cells specified, the extra cells lag behind, potentially through different mechanisms than de-adhesion.

Philipps-Universität - 35037 Marburg

Dr. William Bement
Monitoring Editor

Dr. Tim Fessenden
Scientific Editor
Journal of Cell Biology

Institute for Physiology and
Pathophysiology
Dept. of Molecular Cell
Physiology
Head of Department
Prof. Dr. Sven Bogdan
Philipps-University Marburg
Emil-Mannkopff-Str. 2
35037 Marburg
Germany
Tel.: ++49 6421 28 26 816
Fax: ++49 6421 28 68 690
E-Mail: sven.bogdan@staff.uni-
marburg.de

Marburg, 19th July 2024

JCB manuscript #202310153 – manuscript revision

Dear Dr. Bement, dear Dr. Fessenden,

thank you very much again for giving us the opportunity to revise our manuscript entitled “CYRI controls epidermal wound closure and cohesion of invasive border cell cluster in *Drosophila*” (JCB manuscript #202310153) and the reviewers for their careful and very constructive comments.

Let me briefly highlight the **most important points** that we have addressed before I provide a detailed point-to-point- response to the referees' comments.

1) First, we significantly improved our quantitative analysis of wound closure by increasing the number of previous experiments from nine 60-minute live cell recordings to 15 per genotype. This means an analysis of more than 4000 min, i.e. **more than 66 hours of additional video material**.

2) Second, we performed **rescue experiments** (CYRI-WT versus CYRI-R163/164D) and **single *cyri* RNAi experiments** that confirmed a specific and cell-autonomous of

function of CYRI in epidermal wound closure. Additional **double RNAi experiments** further showed an epistatic relationship between *cyri* and *wave* in wound closure.

3) Third, we established a new GFP-based sensor, termed MBT-GFP. This sensor includes the Cdc42- and Rac-interactive binding (CRIB) domain of the p21-activated kinase *mushroom bodies tiny* (Mbt) fused to EGFP (Melzig et al., 1998). GST-pull down experiments confirmed its specific binding to GTP-loaded *Drosophila* Rac1 and Rac2. MBT-GFP also strongly bound Cdc42, but this binding was, however, independent of the nucleotide-binding status of Cdc42. *In vivo*, the reporter nicely marks the increased lamellipodial protrusions of epithelial cells at the wound margin depleted for *cyri*.

4) Fourth, new time-lapse microscopy data of **stable S2 CYRI-GFP knock-in single-cell clones** showed for the first time that endogenous CYRI protein indeed localizes at leading pseudopods, thus, behaves like a ‘local inhibitor’ of WRC, as previously suggested based on cell transfection experiments with ectopically expressed CYRI constructs.

5) Finally, new quantitative confocal microscopy analysis of RNAi and overexpression experiments showed that **CYRI also controls cell-autonomously** not only border cell cohesion but also WAVE-driven border cell cluster motility.

As you will see, we have also addressed **all other reviewer’s comments** and provide a set of new experiments including in three changed main figures, one new main figure, two new supplementary figures and three supplementary movies.

Below I provide a detailed response to the referees’ comments.

With best regards
Sven Bogdan

Reviewer #1 (Comments to the Authors (Required)):

My overall assessment is that this is a solid paper that covers a lot of ground, but it is suited for a more specialist journal than JCB. There is a lot of rigorous work presented, geared generally to addressing the question, what does CYRI do? Many of the experiments and results confirm previous findings (acknowledged by the authors), which is reassuring, including the data in Figs. 1-3.

Previous pioneering work on the function of CYRI was limited almost exclusively to *in vitro* experiments with cultured mammalian cells. The role of CYRI proteins *in vivo* in healthy tissues was unclear. Thus, in our study we analyzed CYRI function for the first-time in a physiological context at organismal level. Since the CG32066 gene has not yet been characterized in flies, we first had to confirm that this annotated unknown gene is indeed a functional ortholog of mammalian CYRI A/B proteins, showing that *Drosophila* CG32066 encodes a Rac-binding proteins that controls lamellipodial cell spreading in cultured cells. This necessary analysis comprises only the first part of our analysis, the second and the third part of our study further provided new insights into the *in vivo* role of CYRI in a physiological and pathophysiological context of wound healing tissue dynamics and invasive collective cell behavior.

As specifically suggested by reviewer #2, we further improved our data for the first part of our study. First, we have now repeated our pull-down assays with recombinant CYRI protein in three independent experiments and confirmed that *Drosophila* CYRI preferentially binds activated Rac (see also new figure 1D, E). More importantly, new time-lapse microscopy data of a clonally selected stable S2 CYRI-GFP knock-in cell line showed for the first time that endogenous CYRI protein localizes at leading pseudopods, thus, behaves indeed like a 'local inhibitor' of WRC, as previously suggested based on cell transfection experiments with ectopically expressed CYRI constructs. These new data are included in new figure 2A-C.

The main contribution of the manuscript is in Fig. 5, where the CRYI wound healing data is presented, and in Fig. 4 containing the WRC data, a companion to Fig. 5. The data on border cell migration in Figs. 6 and 7, although interesting and generally confirming the function of CRYI in collective epithelial cell migrations, points in a new

direction - integrin regulation by CRYI - which is not analyzed in any depth. I wish the authors had instead focused on either wound healing or border cell migration to understand one phenomenon more fully.

We agree with the reviewer that the second and third part of our study provide the most interesting, new findings about the role of CYRI in controlling collective epithelial cell migration. In the second part of the revised manuscript, we provided more than 66 hours of additional video material, including rescue and single/double RNAi experiments that confirmed a cell-autonomous function of CYRI in controlling epidermal wound closure, included in new figures 4F, F'; figure 5 E, E'; new supplementary figure S2. These data also include live-imaging experiments with a new GFP-based sensor for active Rac1 and Rac2 (MBT-GFP). GST-pull down experiments confirmed its specific binding to GTP-loaded *Drosophila* Rac1 and Rac2. MBT-GFP also strongly bound Cdc42, but this binding was, however, independent of the nucleotide-binding status of Cdc42 (supplementary figure S3A). In vivo, the reporter nicely marks the increased lamellipodial protrusions of epithelial cells at the wound margin depleted for cyri (figure 5F, G). In our revised manuscript, we also demonstrated that loss of CYRI function accelerates wound closure, not only in single-cell wounds but also in larger, multicellular epidermal wounds (new supplementary figure S2B-D).

For the third part of our revised manuscript, new quantitative confocal microscopy analysis of RNAi and overexpression experiments showed that CYRI controls cell-autonomously not only border cell cohesion but also WAVE-driven cluster motility. These new data include cell-specific RNAi and overexpression of CYRI and WAVE experiments either in outer border cells or in polar cells (see new figure 7).

The paper has a rocky start with way too much molecular detail in the first paragraph, and I strongly suggest that the authors rewrite the introduction, focusing on the bigger picture of actin protrusions, when they are needed in individual cells and in epithelia, and feed-forward and feedback loops that control them. If the molecular details that are currently presented in the first paragraph of the introduction are necessary, they need an accompanying figure.

We have significantly revised the introduction by presenting the most important findings in a clearer form, in particular we have focused more on the formation of actin protrusions and feedback loops that control them.

Reviewer #2 (Comments to the Authors (Required)):

The manuscript by Rötte and colleagues describes the characterisation of CYFIP in *Drosophila* epidermal and epithelial tissues to better understand its function in cell dynamics *in vivo*.

This study builds on recent work from the Machesky lab, which has reported that CYFIP-related Rac1 interactor (CYRI) proteins compete with the WAVE regulatory complex (WRC) for interaction with Rac1 in a feedback loop, regulating lamellipodia dynamics. However, the physiological role of CYRI proteins in normal tissues had not been established. This study adds to the knowledge about CYRI by addressing this issue in a well-studied and tractable model system.

Although not essential for viability, this study finds that in CYRI controls lamellipodia spread in isolated macrophages, wound healing in epidermal cells and collective cell migration in migratory epithelial cells of the ovary. The genetic data and molecular analysis presented is consistent with *Drosophila* CYRI acting as a regulator of WRC, although some uncertainty remains about its precise mode of action.

Overall, this manuscript presents new findings on the physiological role of a CYRI protein, which has previously been described as a Rac1-binding regulator of lamellipodia and macropinocytic events. The manuscript is generally well-written. Most of the conclusions are supported by the data presented.

We thank the reviewer for the positive comments.

However, some aspects appear more preliminary due to lack of quantification and incomplete statistical analyses. The following weaknesses should be addressed:

1. Sample sizes are not always stated, statistical analyses are missing or are not appropriate, and the variation for each genotype is not clearly displayed in figures or reported in the text. The authors should clarify how many different times independent experiments were performed, and how many flies were analysed per experiment. At least three independent experiments should be carried out for each genotype.

We apologize that sample sizes are not always stated in figure legends. In the revised manuscript, we carried out at least three independent experiments for each genotype or experiments and clearly displayed sample sizes and statistical analyses in figures or figure legends. Rather, we have significantly improved our quantitative analysis of wound closure by increasing the number of experiments from nine 60-minute live cell recordings to 15 per genotype. Including our new *cyri* RNAi experiments, this means an analysis of more than 4000 minutes, i.e. more than 66 hours of additional video material. Additional data including the rescue and single/double RNAi experiments confirmed a cell-autonomous function of CYRI in controlling epidermal wound closure, included in new quantifications depicted in figure 4F, F'; figure 5 E, E'; new supplementary figure S2.

2. When comparing more than two conditions, the authors should use either non-parametric, Kruskal-Wallis test with Dunn's Multiple Comparison test or one-way ANOVA with Tukey Multiple Comparison test to reveal statistically significant differences between datasets throughout, depending on whether the data are distributed normally. Mann-Whitney and T tests are inappropriate when there are more than two independent samples. It is critical to measure the variation within and between experiments to identify significant differences between different genotypes.

As mentioned above we revised and improved our statistical analyses. Again, we now quantified 15 instead of previously 9 experiments per genotype (60-minute live cell recordings). In detail, one-way ANOVA is an analysis of variance in which there is only one independent variable. Instead, we now used the two-way ANOVA analysis that allows to compare the mean of the WT column against the other columns since we have multiple subcolumns (n=12-18). We also wanted to analyze if each time point (row) of each genotype significantly differs against the WT. Therefore, we corrected after Dunnett as our sub-columns are repeated measurements.

3. While the data in Figure 1 suggest CG32055/*Drosophila* CYRI can interact with activated Rac1, the Western blots and immunofluorescent images in Figure 1 need to be quantified, along with biological repeats, and the data subjected to appropriate statistical analysis. Further, the data presented is based entirely on overexpression, both in cells (Fig.1D-E) and in imaginal discs (Fig.1F-G). If possible, the authors

should attempt to demonstrate an interaction between endogenous proteins takes place, to increase confidence that CYRI is a Rac1 binder in vivo.

As suggested by the reviewer, we have now repeated our pull-down assays with recombinant GST-CYRI protein in three independent experiments and could confirm that *Drosophila* CYRI preferentially binds activated Rac1-V12 variant compared to Rac1-N17 (see also new figure 1D and quantification of three independent pull-down experiments in figure 1E). In addition, we now performed similar pull-down experiments using GST-tagged wild type Rac1 (Rac-WT) and constitutively active Rac, RacQ61L, incubated either with cell lysates expressing HA-tagged wild type CYRI (HA-CYRI-WT) or the mutant HA-tagged CYRI^{R163/164D} (supplementary figure S1B). CYRI-WT interacted more strongly with RacQ61L compared to Rac-WT as previously shown for mammalian CYRI-B (Fort et al., 2018). Again, mutations of key arginines to aspartic acid in CYRI abrogated this interaction (supplementary figure S1B).

We have also tried several times to show binding of endogenous CYRI with recombinant Rac1 and Rac2, but failed. We also tried to test whether in the absence of WAVE an increased amount of “free” endogenous CYRI protein binds Rac. Here, we used a stable *Drosophila* S2 cell line in which WAVE was deleted by using CRISPR/Cas9-mediated genome editing. However, we could not detect a significantly increased binding of endogenous CYRI to recombinant Rac.

4. The experiments examining the subcellular localisation and effect of CYRI on lamellipodial protrusions are somewhat preliminary. The anti-CYRI staining in Fig 2A is rather weak. Although the antibody appears to specifically detect the untagged protein on Western blots (Figures 2B and 3B) and a proportion of cells ectopically expressing CYRI (Fig.2F-G), it would be useful to show that the immunofluorescent signals in Fig.2A are lost after RNAi-knockdown. Does CYRI colocalise with F-actin in Fig.2A, or with Scar/WAVE?

We thank the reviewer for these suggestions. We agree that the anti-CYRI antibody specifically detects the endogenous protein in the western blots, but the specificity of the immunofluorescence signals in cultured S2 cells were not as convincing. We therefore performed immunofluorescence staining of *Drosophila* immune cells, now

both in wild type and *cyri* mutant macrophages isolated from third instar larvae. These new data show that the antibody did not detect specifically the endogenous CYRI protein. Both intensity and localization of the immunofluorescence signals in wild-type and mutant cells are comparable as shown in new supplementary figure S1C, D.

Instead, we further analyzed the more promising localization and dynamics of endogenously GFP-labeled CYRI protein. As already noticed by this reviewer, this C-terminal tagged CYRI-GFP protein localized to some extent to the cell periphery in fixed cells co-stained with phalloidin. We now clonally selected, stable S2 cells and confirmed GFP expression in growing culture by repeated FACS sorting and western blot experiments. We indeed confirmed its increased localization at the cell periphery. Our new live-cell confocal analysis shows a prominent dynamic enrichment at lamellipodial tips. These new data further support the model that CYRI behaves like a 'local inhibitor' of WRC at lamellipodial tips by restricting Rac1's activity at the cell membrane. We also frequently observed a localization of CYRI-GFP to macropinocytic structures suggesting a conserved function in macropinocytosis as recently found in mammalian cell culture. We included these new important data in a new figure 2E as well as a supplementary movie M2.

5. Noticeably, the distribution of the CYRI antibody stain is different from the distribution of GFP-CYRI in Fig.2D', where there appears to be accumulation in the nucleus and some signal at the cell periphery. This might suggest that the distribution of the GFP-CYRI protein has been affected by the GFP tag - failure to detect the protein on a blot at least suggests a change in accessibility of the anti-CYRI epitope. Although I appreciate the effort to make the CRISPR line, I'm not sure what it adds unless the distribution can be shown to faithfully recapitulate the endogenous untagged protein.

As mentioned above, the antibody did not detect specifically the endogenous CYRI protein in immunostainings. We now selected stable GFP *knock-in* S2 cell clones. In new Western blot experiments, we now confirmed the expression of this endogenously GFP-tagged CYRI protein using both an anti-GFP and an anti-CYRI antibody. In both cases, we were able to detect a protein of the expected size of 65 KDa (see also new figure 2A). Since the GFP fluorescence is very low (endogenous levels), it was very challenging to image these cells *in vivo*. However, we could

indeed confirm its increased localization at cell periphery and some accumulation in the nucleus. New live-cell confocal analysis confirms a prominent dynamic enrichment at lamellipodial tips. These new data further support the current model that CYRI behaves like a 'local inhibitor' of WRC at lamellipodial tips by restricting Rac1's activity at the cell membrane. We also observed frequently a localization of CYRI-GFP to macropinocytic structures suggesting a conserved function in macropinocytosis as recently found in mammalian cell culture. We included these new data in the new figure 2E as well as a supplementary movie M2.

Since adding a GFP-tag to either end of CYRI-B interfered with its function upon ectopic expression in mammalian cell culture (see Fort et al., 2018), we further ruled out possible effects of the C-terminal tag on the functionality of the endogenous CYRI protein by quantifying possible changes in cell spreading. However, we did not find significant changes in lamellipodia spreading of CYRI-GFP cells compared to S2 Cas9 control cells (see also new figure 2B).

6. The authors show that ectopic expression of the CYRI protein changes the cell morphology in a proportion of S2 cells (Fig2F-G). Some cells appear unaffected, but this is probably related to CYRI expression level/transfection efficiency. It would be useful to quantify this relationship. Furthermore, the quantification in Fig2. is impossible to interpret without knowing the number of biological repeats and distribution of the data (see point 1).

We apologize that sample sizes are not always stated in figures and/or figure legends. In the revised manuscript we clearly displayed sample sizes and statistical analyses in the figure or figure legend. We always carried out at least three independent experiments for each genotype or experiment. In Figure 2I, we only quantified transfected cells (overall transfection efficiency about 60-70%) marked either by GFP expression (control) or stained for ectopic CYRI recognized by the anti-CYRI antibody. For CYRI-WT overexpression about 80% of all transfected (identified by anti-CYRI staining) showed a spiky phenotype, whereas 20% had a normal pancake-like morphology despite ectopic overexpression which was confirmed by immunostaining. In contrast, overexpression of CYRI-R163/164D resulted in only about 15% spiky cells similar to controls transfected with a GFP construct. By contrast, 85 % of cells expressing CYRI-R163/164D with confirmed expression show a wild type cell morphology.

7. To strengthen the implied mechanism by which ectopic CYRI proteins are acting, it would be helpful to examine whether ectopic expression of CYRI mutant or CYRI RNAi knockdown results in enlarged or broad lamellipodia as it does in human cells. This would also strengthen the findings in Fig.3. Is the recruitment of Scar/WAVE affected as it is in human cells? Furthermore, is ectopic CYRI wt capable of suppressing any Rac1-hyperactivation phenotypes, including increased cell circularity?

As suggested by the reviewer, we tested two independent transgenic *cyri*-RNAi fly lines from VDRC and quantified changes in cell size upon macrophage-specific expressing using the *hmlP2A*-Gal4 driver. Both RNAi lines (#1 and #2) were indeed functional resulting into increased macrophage cell size. The transgenic RNAi #1 line was stronger than RNAi line #2. Additional quantification further revealed an increased immunofluorescent anti-WAVE intensity at the leading edge in cell depleted for CYRI (*cyri* RNA #1). Both new data sets are included in the new figure 3C-G. Both *cyri*-RNAi fly lines were also tested in epidermal wound closure and border cell (BC) migration. Epidermal and BC expression of both RNAi fly lines phenocopied *cyri* loss-of-function (see new figure 7 and supplementary figure S2A).

We also tried to co-overexpress wild type CYRI together with Rac. However, this co-expression results in larval lethality with many melanized inclusion bodies that did not allow further phenotypic analysis.

8. The conclusions in Figure 3 seem well supported, including genetic evidence from transheterozygous and hemizygous mutant animals that loss of CYRI in flies is not essential for viability. Although only found in one genotype (*cyri*[d11]/Df), the cell spreading phenotype looks convincing and is reminiscent of the *cryi*-b knockdown phenotype in mammalian cells. Again, staining of these cells with Scar/WAVE proteins to examine its recruitment to lamellipodia would help to confirm the underlying mechanism.

We further validated our quantitative analysis of the spreading phenotype of isolated macrophages by using the *cyri* null deficiency, Df(3L)ED4457. This deficiency was already used for all mutant analysis in wound closure and border cell cluster but not

for analysis of the spreading phenotype of isolated macrophages. In the original manuscript submission, we analyzed transheterozygous combinations with the third chromosomal *cyri* null deficiency Df(3L)BSC439. In the revised manuscript, we now included the macrophage data obtained from *cyri*/ Df(3L)ED4457 transheterozygous animals. These new data confirmed that *cyri* Δ 2 is a hypomorphic allele, rather than a null allele as *cyri* Δ 11. However, homozygous and transheterozygous *cyri* Δ 2/ Df(3L)ED4457 macrophages also showed an increased cell spread although weaker compared to transheterozygous *cyri* Δ 11/ Df(3L)ED4457 macrophages (new supplementary Figure S1E-I).

More importantly, we now tested two independent transgenic *cyri*-RNAi fly lines from VDRC and quantified changes in cell size upon macrophage-specific expressing using the *hmlP2A*-Gal4 driver. Both RNAi lines were indeed functional resulting into increased macrophage cell size. Additional quantification further revealed an increased immunofluorescent anti-WAVE intensity (highly specific affinity-purified antibody) at the leading edge of macrophages depleted for CYRI. Both new data sets are included in figure 3C-E, quantification in figure 3F, G). Both *cyri* RNAi fly lines were also tested in epidermal wound closure and border cell migration. Both RNAi fly lines phenocopied *cyri* loss-of-function in wound closure and border cell cluster integrity (see new figure 7 and supplementary figure S2A). By contrast, overexpression of CYRI resulted in smaller, spiky cells with decreased immunofluorescent anti-WAVE intensity (see also figure 3E, quantification in figure 3G).

9. In the abstract, the authors state that *cyri* controls both lamellipodia spread and protrusion dynamics. However, as far as I can see there is no analysis of protrusion dynamics. It would be useful to show effects on dynamics in any one of the experimental systems.

We agree with the reviewer. As mentioned above, we now analyzed lamellipodial protrusions *in vivo* during wound closure using a GFP-based Rac/Cdc42 sensor (MBT-CRIB-GFP). *In vivo*, the reporter nicely marks the increased lamellipodial protrusions of epithelial cells at the wound margin depleted for *cyri* (Figure 5F, G, supplementary movie M6).

10. Experiments analysing the effect of CYRI and Rac-WRC-Arp2/3 pathway during wound closure are informative and support the proposed model. However, it looks like the control in Fig.4F-G is the same as the one in Fig.5E-F. I can understand that the data have been presented in this way for clarity. However, this issue needs to be clarified in the figure legends and the statistical tests should be run across all genotypes using the appropriate multiple comparison test to take account of variance (see point 2).

We apologize for any confusion. In fact, we have shown the same wild-type controls in both figures, as the data belong to the same dataset and were originally presented in one graph. In detail, the video sequences of the first 15 minutes of the original 60-minute live videos, which document the entire course of the wound closure, were used to analyze the lamellipodia width. To further improve our quantifications, we now further increased the sample size for each genotype from initially $n=9$ to $n=12-18$ independent wound experiments. In the revised manuscript, we have now clearly presented the sample sizes and statistical analyses in the figure or figure legend.

11. Switching to a model of collective migration, the authors show that *cyri[d11]/Df* flies display a border cell migration defect, which may explain an observed reduction in fecundity. It is impossible to tell from these experiments whether the requirement for *cyri* is in just the migratory outer border cells or whether *cyri* also plays a role in non-migratory polar cells. Furthermore, given the reliance on just *cyri[d11]/Df* for phenotypic information in this context, it would be reassuring to phenocopy the effect with *cyri* RNAi or look for complementation with *cyri* wt overexpression specifically in the outer border cells.

We thank the reviewer for this suggestion. In the revised manuscript, we overexpressed the CYRI and *cyri* RNAi transgenes differentially in outer border cells (C306-Gal4) and in non-migratory polar cells (*upd*-Gal4). RNAi-mediated depletion in border cells using the *c306*-Gal4 driver line also resulted in prominent lagging border cells (Figure 7A). Expression of the same both RNAi transgenes under the control of the *upd*-Gal4 driver (E132-Gal4), which is exclusively expressed in polar cells, did not result in any significant phenotype suggesting that CYRI function is only needed migratory outer border cells (Figure 7A'). Interestingly, overexpression of wild type CYRI but not Rac-binding deficient CYRIR162/163D variant under the control of the *c306*-Gal4 driver did not result in cohesion defects but rather in migration defects of

border cells cluster (Figure 7D; quantification in figure 7A). Both, delayed migration and cohesion defects of border cell cluster were also seen upon RNAi-mediated depletion of WAVE (Figure 7E; quantification in figure 7A). Given that Rac activity must be tightly regulated in both leader and follower cells, increased pools of activated Rac might affect not only migration but also cohesion of border cell cluster (Campanale et al., 2022). Supporting this notion, we found that overexpression of a membrane-tethered activated WAVE variant (WAVEMyr; (Stephan et al., 2011) in outer border cells, phenocopied loss of *cyri* function, thus resulting in significant reduction of cluster cohesion (Supplementary figure 4B-D; quantification in 7A).

12. It is interesting that membrane tethered activated WAVE had a weaker effect than *cyri* loss of function, leading the authors to test the involvement of *cyri* in regulating cluster cohesion via integrin-mediated adhesion. Le et al (2021), previously reported that CYRI proteins affect integrin trafficking, so perhaps differences in integrin distribution is not unexpected. The parallels with this study should be mentioned in the text. The authors should test whether this role is Rac dependent, by testing for a lack of complementation with their UAS-*cyri* mutant, compared to the wild type transgene.

As suggested, we now mentioned the parallels in the revised manuscript (see discussion). Experimentally, we overexpressed both wild type CYRI and mutant CYRI deficient for Rac binding. Interestingly, we found that CYRI-WT overexpression under the c306-Gal4 driver caused neither lagging border cell phenotype nor a mislocalization of β -integrin (new figure 8G). However, CYRI-WT overexpression rather resulted in a prominent delay of border cell migration, a phenotype which resembles a RNAi-mediated knock-down of WAVE function (see also new figure 7A-D). By contrast, overexpression of the CYRIR162/163D variant did not result in any phenotype, neither using the c306-Gal4 nor the E132-Gal4 driver (see also new figure 7A, A').

Minor issues

Line 192: some general information about the gene, e.g. the cytological location, doesn't seem particularly relevant and can be removed.

We removed this passage from the text.

Line 357-361: this is a long sentence interrupted with a full-stop.

We modified the text to improve readability.

Line 375: The authors should clarify the following statement, which could seem counterintuitive to the reader "As in cultured mammalian cells, *Drosophila* CYRI broadly localized in the cytoplasm. Thus, CYRI behaves like a 'local inhibitor' of WRC at lamellipodial tips by restricting Rac1's activity at the cell membrane."

We agree with the reviewer and we modified the text accordingly.

Reviewer #3 (Comments to the Authors (Required)):

Summary

This manuscript from Rotte et al characterizes the *Drosophila* ortholog of the mammalian proteins, CYRI-A/B, which can regulate actin dynamics through altering WAVE and Rac activity. This gene has not been previously characterized in flies, so the paper makes a good contribution and provides foundation for further study. Rotte et al examine the expression of CYRI in S2 cells in culture, and generate tools to characterize it further, specifically they made an antibody, generated two mutant alleles via CRISPR, and two inducible overexpression strains, one wild type and one mutant. CYRI interacted with Rac1 in pull-down assays and physically in wing disc. Cyri overexpression collapsed lamellopodia in cell culture, which phenocopies loss of wave function, and this required Rac1-binding-residues in CYRI. The overexpression phenotypes resemble those for loss of wave components in an epithelial wound healing assay. Interestingly, Cyri gain or loss of function could control the speed at which the wound closes. Additional cyri mutant defects are described for ovarian border cell migration.

Generally, the manuscript is very clear and well written, and the data are presented well. However, some of the data is circumstantial and not definitive, additional experiments are needed to pin down what is happening with the cyri mutants and to connect CYRI and a functional mechanism. I provide more details below.

We thank the reviewer for the positive comments.

Major comments

The authors generate two loss of function mutant alleles, and determine that delta2 is a hypomorph and delta11 is a null. They test these in various combinations and in trans to deficiencies. However, they do not rescue the mutant defects by putting the gene back, which is needed to confirm the phenotypes they see are due to the *cyri* mutations. This is especially important since they acknowledge there seems to be an off-target hit that makes the delta 11 allele homozygous lethal. I understand that overexpression has defects on its own, but it could be expressed at low levels.

As suggested by the reviewer, we performed rescue experiments with CYRI-WT in comparison to the CYRI-R163/164D mutant. For this purpose, we generated new UAS-CYRI-WT and UAS-CYRI^{R163/164D} transgenes on the second chromosome necessary for rescue experiments. Using new recombined fly mutant stocks, we now could show that re-expression of a wild type CYRI but not Rac-binding deficient CYRI-R162/163D variant rescued the *cyri* loss of function phenotype resulting in lamellipodia sizes similar to wild type epidermal cells (new Figure 5E').

In wound closure, gain of CYRI can inhibit or promote cell dynamics in ways that are similar to knockdown of WAVE or Arp2/3. But this correlative data does not necessarily mean they work in the same pathway. The argument would be more convincing if they did genetic interaction tests here, or tried to overcome knockdown of wave by genetic reduction of *cyri*, for example. These or similar experiments are needed to substantiate the claim in the abstract that CYRI is a "potent WRC regulator."

We thank the reviewer for this suggestion. We now provide further evidence that model that CYRI behaves like a 'local inhibitor' of WRC at lamellipodial tips. First, we now provided new macrophage-specific RNAi and overexpression experiments that further confirmed that *Drosophila* CYRI affects lamellipodial spreading by regulating the level of endogenous WAVE protein at the cell's leading edge (see new figure 3C-D).

Second, simultaneous RNAi-mediated suppression of *cyri* and *wave* still resulted in lamellipodial protrusion defects similar to wave RNAi depletion alone upon wounding

(supplementary Figure S2A). Thus, this further suggests that CYRI indeed acts through WAVE in regulating lamellipodial protrusions during wound closure.

In addition, cohesion defects observed in *cyri* mutants could partly be due to overactivation of WAVE. Supporting this notion, we observed a significant decrease of lagging border cells and delayed border cell clusters in egg chambers depleted for both, *cyri* and *wave* (new Figure 7A). Similarly, we observed that removal of one copy *wave* in *cyri* mutant background reduced the lagging border phenotype, although the difference was not significant (Supplementary figure S4E).

The authors use *cyri* mutant egg chambers to study border cell migration, which means that both border cells and the nurse cell substrates would be affected. To interpret this phenotype clearly, one needs to know in which cell type(s) the gene is acting. The authors may be able to use their antibody to see where in the egg chamber CYRI is expressed, or they could do cell autonomous knockdown or rescue experiments in some cells using the lines they have.

Again, we thank the reviewer for this suggestion. As mentioned before, we overexpressed the CYRI and *cyri* RNAi transgenes differentially in outer border cells (c306-Gal4) and in non-migratory polar cells (upd-Gal4). RNAi-mediated depletion in border cells using the c306-Gal4 driver line resulted in prominent lagging border cells (Figure 7A). Expression of the same both RNAi transgenes under the control of the upd-Gal4 driver (E132-Gal4), which is exclusively expressed in polar cells, did not result in any significant phenotype suggesting that CYRI function is only needed migratory outer border cells (Figure 7A'). Thus, these data show a cell-autonomous function of CYRI in border cells. Interestingly, overexpression of wild type CYRI but not Rac-binding deficient CYRIR162/163D variant under the control of the c306-Gal4 driver did not result in cohesion defects but rather in migration defects of border cells cluster (Figure 7C; quantification in figure 7A). Both, delayed migration and cohesion defects of border cell cluster were also seen upon RNAi-mediated depletion of WAVE (Figure 7E; quantification in figure 7A). Given that Rac activity must be tightly regulated in both leader and follower cells, increased pools of activated Rac might affect not only migration but also cohesion of border cell cluster (Campanale et al., 2022). Supporting this notion, we found that overexpression of a membrane-tethered activated WAVE variant (WAVEMyr; Stephan et al., 2011) in outer border cells,

phenocopied loss of *cyri* function, thus resulting in significant reduction of cluster cohesion (Supplementary figure 4B-D; quantification in 7A).

We also tried to use our anti-CYRI antibody to describe the localization of endogenous CYRI protein in egg chambers. However, we found that the antibody did not detect specifically the endogenous CYRI protein neither in macrophages (new supplementary figure S1C, D) nor in egg chambers. Both intensity and localization of the immunofluorescence signals in wild-type and mutants are comparable. In addition, we also tested available CYRI FlyPhos and MiMiC lines to shine light on the localization and expression pattern of CYRI in egg chambers but all lines did not give a specific signal.

It would be interesting to see if CYRI overexpression has an effect in border cells and if the R163/164D mutant could rescue the loss of function.

As mentioned before, we indeed found that overexpression of wild type CYRI but not Rac-binding deficient CYRIR162/163D variant under the control of the c306-Gal4 driver did not result in cohesion defects but rather in migration defects of border cells cluster (Figure 7C; quantification in figure 7A). Both, delayed migration and cohesion defects of border cell cluster were also seen upon RNAi-mediated depletion of WAVE (Figure 7D; quantification in figure 7A). Given that Rac activity must be tightly regulated in both leader and follower cells, increased pools of activated Rac might affect not only migration but also cohesion of border cell cluster (Campanale et al., 2022).

As mentioned above, we did rescue experiments with CYRI-WT and the CYRI-R163/164D variant in wound closure. We now showed that re-expression of a wild type CYRI but not Rac-binding deficient CYRIR162/163D variant rescued the *cyri* loss of function phenotype resulting in lamellipodia sizes similar to wild type epidermal cells (new Figure 5E').

In egg chambers, we tried to suppress the *cyri* RNAi-induced phenotype by co-overexpression of CYRI. Indeed, we observed that both the delayed border cell phenotype evoked by CYRI overexpression and the *cyri* RNAi-induced lagging border cell phenotype could be completely suppressed (Figure 7A).

In addition, it would again be helpful to look for genetic interactions between mutants for CYRI and the predicted interacting proteins, eg, wave or Rac to implicate them in functional interactions.

We indeed performed double RNAi experiments in epidermal wound closure and border cell clusters. In epidermal wounds, we did not observe a suppression of the wave RNAi phenotype when we simultaneously reduced CYRI function by RNAi suggesting that CYRI indeed acts through WAVE in regulating lamellipodial protrusions during wound closure (Figure S2A).

However, in egg chambers, we observed a significant decrease of lagging border cells when we depleted for both, *cyri* and *wave* compared to single RNAi experiments (new figure 7A). Similarly, we observed that removal of one copy *wave* in *cyri* mutant background reduced the lagging border phenotype, although the difference was not significant (supplementary figure S4E). Thus, we found no simple epistatic relationship between *cyri* and *wave* but a more complex interaction that could also reflect a possible role of CYRI in Rac1-dependent β -integrin trafficking required in border cell cohesion.

I found the changes to integrin staining subtle and hard to see. How was this measured for figure 7F?

We provided a detailed description in the M&M to explain in more detail how the image analysis was performed. For anti-mys intensity measurements, lines were drawn along three membranes between outer border cells (“arms”) and the membrane where polar and border cells connect (apical ring). The average intensity along the arms was divided by the intensity along the apical ring. The ratio was plotted and statistical significance was determined using Mann-Whitney-test.

It would be better to normalize the levels against something not likely to change (eg, membrane GFP), or even more convincing to see compensatory changes in integrin when *cyri* was overexpressed in border cells.

As suggested, we performed these overexpression experiments. Overexpression of CYRI under the control of the c306-Gal4 driver line did not result in significant changes in bPS-integrin localization (see quantification in new Figure 8G).

Minor comments

The introduction has a lot of jargon and is a bit challenging to follow without knowing a fair amount about WAVE regulatory complex and Rac already.

We modified the introduction to improve readability.

The authors made an antibody, but it doesn't detect the GFP-fusion protein, which is odd. Maybe they can explain this?

In new Western blot experiments, we now confirmed the expression of this endogenously GFP-tagged CYRI protein using both an anti-GFP and an anti-CYRI antibody. In both cases, we were able to detect a protein of the expected size of 65 KDa (see also new figure 2A). Since the GFP fluorescence is very low (endogenous levels), it was very challenging to image these cells *in vivo*. However, we could indeed confirm its increased localization at cell periphery and some accumulation in the nucleus. New live-cell confocal analysis confirms a prominent dynamic enrichment at lamellipodial tips. These new data further support the current model that CYRI behaves like a 'local inhibitor' of WRC at lamellipodial tips by restricting Rac1's activity at the cell membrane. We also observed frequently a localization of CYRI-GFP to macropinocytic structures suggesting a conserved function in macropinocytosis as recently found in mammalian cell culture. We included these new data in the new figure 2 as well as a supplementary movie M2.

Can the authors use the expression analysis tools (antibody or GFP) to look where CYRI is expressed in pupal epithelia? Does it localize to the cortex of cells under normal conditions? Does it localize to the actinomyosin ring? These data may also help make a clearer case for CYRI function.

Our anti-CYRI antibody specifically detects the endogenous protein in the western blots, but did not detect specifically the endogenous CYRI protein in immunostainings, neither in macrophages (new supplementary figure S1C-D) nor in egg chambers or abdominal epidermis. Both intensity and localization of the immunofluorescence signals in wild-type and mutants are comparable. In addition, we also tested available CYRI FlyFos and MiMiC lines to shine light on the

localization and expression pattern of CYRI in egg chambers but all lines did not give a specific signal. We only found one Gal4 enhancer trap line inserted into the CG32066 locus, termed P(GawB)CG32066^{NP2400} (see also flybase) that drives a fluorescent UAS reporter (e.g. UAS-GFP/RFP) specifically in the pupal epidermis suggesting that endogenous CYRI is indeed expressed in epidermal cells, as also found in single-cell transcriptomic fly atlas (Li et al., 2022).

Line 242/3 seems like a word is missing? "...compensate progressively (for) initial wound closure..."

We corrected the text accordingly.

Line 278 should be 6D not 5D

We corrected this mistake.

Figure panel 2H should have n values.

We included n-values in figures and figure legends throughout the manuscript wherever appropriate.

How does the R163/164D mutant CYRI affect border cells?

In border cells, overexpression of the CYRIR162/163D mutant variant did not result in any phenotype, neither using the c306-Gal4 nor the E132-Gal4 driver (see also new figure 7A, A').

The text states (line 286/7) "mutant border cell clusters that reached nurse cell-oocyte boundary often contained less migratory border cells (Figure 6G3). However, this was not quantified. The activated wave phenotype in border cells appears to be much stronger in terms of fewer cells overall. These border cells should be quantified in both cases. It is an important point since in mutants that have additional border cells specified, the extra cells lag behind, potentially through different mechanisms than de-adhesion.

We thank the reviewer for the suggestion. We indeed quantified the total cell number of border cells, however it was not changed (new supplementary figure S4A).

September 5, 2024

RE: JCB Manuscript #202310153R-A

Prof. Sven Bogdan
University of Marburg
Institute of Physiology and Pathophysiology
Marburg 35037
Germany

Dear Prof. Bogdan:

Thank you for submitting your revised manuscript entitled "CYRI controls epidermal wound closure and cohesion of invasive border cell cluster in *Drosophila*". Reviewers are satisfied with the changes in place but make a few minor requests to improve the text and clarity of the data. In addition, we have evaluated the new data included in Figure 5 and we feel these observations are appropriate to include in the manuscript without reviewer input. We would be happy to publish your paper in JCB pending these reviewer requests and final revisions necessary to meet our formatting guidelines (see details below).

A. MANUSCRIPT ORGANIZATION AND FORMATTING:

Full guidelines are available on our Instructions for Authors page, <http://jcb.rupress.org/submission-guidelines#revised>. Submission of a paper that does not conform to JCB guidelines will delay the acceptance of your manuscript.

1) Text limits: Character count for Articles is < 40,000, not including spaces. Count includes abstract, introduction, results, discussion, and acknowledgments. Count does not include title page, figure legends, materials and methods, references, tables, or supplemental legends.

2) Figures limits: Articles may have up to 10 main figures and 5 supplemental figures/tables.

3) Figure formatting: Scale bars must be present on all microscopy images, including inset magnifications. Molecular weight or nucleic acid size markers must be included on all gel electrophoresis. Please avoid pairing red and green for images and graphs to ensure legibility for color-blind readers. If red and green are paired for images, please ensure that the particular red and green hues used in micrographs are distinctive with any of the colorblind types. If not, please modify colors accordingly or provide separate images of the individual channels.

* Please include molecular weight markers on Fig 1D.

4) Statistical analysis: Error bars on graphic representations of numerical data must be clearly described in the figure legend. The number of independent data points (n) represented in a graph must be indicated in the legend. Statistical methods should be explained in full in the materials and methods. For figures presenting pooled data the statistical measure should be defined in the figure legends. Please also be sure to indicate the statistical tests used in each of your experiments (either in the figure legend itself or in a separate methods section) as well as the parameters of the test (for example, if you ran a t-test, please indicate if it was one- or two-sided, etc.). Also, if you used parametric tests, please indicate if the data distribution was tested for normality (and if so, how). If not, you must state something to the effect that "Data distribution was assumed to be normal but this was not formally tested."

5) Abstract and title: The abstract should be no longer than 160 words and should communicate the significance of the paper for a general audience. The title should be less than 100 characters including spaces. Make the title concise but accessible to a general readership.

6) Materials and methods: Should be comprehensive and not simply reference a previous publication for details on how an experiment was performed. Please provide full descriptions in the text for readers who may not have access to referenced manuscripts. We also provide a report from SciScore and an associate score, which we encourage you to use as a means of evaluating and improving the methods section.

* Please provide full details of CRISPR-induced tagging, isolation of pupal macrophages, and bimolecular fluorescence complementation.

7) Please be sure to provide the sequences for all of your primers/oligos and RNAi constructs in the materials and methods. You must also indicate in the methods the source, species, and catalog numbers (where appropriate) for all of your antibodies. Please also indicate the acquisition and quantification methods for immunoblotting/western blots.

8) Microscope image acquisition: The following information must be provided about the acquisition and processing of images:

- a. Make and model of microscope
- b. Type, magnification, and numerical aperture of the objective lenses
- c. Temperature
- d. Imaging medium
- e. Fluorochromes
- f. Camera make and model
- g. Acquisition software
- h. Any software used for image processing subsequent to data acquisition. Please include details and types of operations involved (e.g., type of deconvolution, 3D reconstitutions, surface or volume rendering, gamma adjustments, etc.).

10) Supplemental materials: There are strict limits on the allowable amount of supplemental data. Articles may have up to 5 supplemental figures. Please also note that tables, like figures, should be provided as individual, editable files. A summary of all supplemental material should appear at the end of the Materials and methods section.

13) ORCID IDs: ORCID IDs are unique identifiers allowing researchers to create a record of their various scholarly contributions in a single place. At resubmission of your final files, please provide an ORCID ID for all authors.

15) A data availability statement is required for all research article submissions. The statement should address all data underlying the research presented in the manuscript. Please visit the JCB instructions for authors for guidelines and examples of statements at (<https://rupress.org/jcb/pages/editorial-policies#data-availability-statement>).

Please note that JCB requires authors to submit Source Data used to generate figures containing gels and Western blots with all revised manuscripts. This Source Data consists of fully uncropped and unprocessed images for each gel/blot displayed in the main and supplemental figures. Since your paper includes cropped gel and/or blot images, please be sure to provide one Source Data file for each figure that contains gels and/or blots along with your revised manuscript files. File names for Source Data figures should be alphanumeric without any spaces or special characters (i.e., SourceDataF#, where F# refers to the associated main figure number or SourceDataFS# for those associated with Supplementary figures). The lanes of the gels/blots should be labeled as they are in the associated figure, the place where cropping was applied should be marked (with a box), and molecular weight/size standards should be labeled wherever possible. Source Data files will be directly linked to specific figures in the published article.

WHEN APPROPRIATE: The source code for all custom computational methods published in JCB must be made freely available as supplemental material hosted at www.jcb.org. Please contact the JCB Editorial Office to find out how to submit your custom macros, code for custom algorithms, etc. Generally, these are provided as raw code in a .txt file or as other file types in a .zip file. Please also include a one-sentence summary of each file in the Online Supplemental Material paragraph of your manuscript.

Journal of Cell Biology now requires a data availability statement for all research article submissions. These statements will be published in the article directly above the Acknowledgments. The statement should address all data underlying the research presented in the manuscript. Please visit the JCB instructions for authors for guidelines and examples of statements at (<https://rupress.org/jcb/pages/editorial-policies#data-availability-statement>).

B. FINAL FILES:

Thank you for your attention to these final processing requirements. Please revise and format the manuscript and upload materials within 7 days. If you need an extension for whatever reason, please let us know and we can work with you to determine a suitable revision period.

Thank you for this interesting contribution, we look forward to publishing your paper in Journal of Cell Biology.

Sincerely,

William Bement
Monitoring Editor
Journal of Cell Biology

Tim Fessenden
Scientific Editor
Journal of Cell Biology

Reviewer #1 (Comments to the Authors (Required)):

The revised manuscript by Rotte et al is improved. I appreciate the inclusion of the new MBT-GFP sensor and the epistasis analysis, as well as the improved clarity. I have only four minor points for clarification:

1. Authors should indicate how many times they replicated the results in Fig. 1F,G.
2. In Fig. 5E and E', and S2A, it is difficult to identify the different colors of lines. The authors should work with this figure to find a way to make it easier to read (increase line thickness, include different shapes, label the lines directly, or some other way)
3. I would suggest moving the data from S2 to the main figure, but it is not necessary if the authors have a reason for preferring this presentation.
4. In the text about border cell migration, the authors write, "we observed that removal of one copy wave in cyri mutant background reduced the lagging border phenotype, although the difference was not significant" (lines 393-395). Because it is not significant, the authors cannot claim this as a difference and must rewrite it to something similar to, "removal of one copy wave in cyri mutant background did not significantly reduce the lagging border phenotype".

Reviewer #3 (Comments to the Authors (Required)):

Rotte et al provide the first characterization of the *Drosophila* ortholog of the mammalian proteins, CYRI-A/B. In the revised manuscript, they have done an impressive job to provide new experimental results, additional live imaging, better quantifications, and some discussion that have strengthened the work. This further supports the main idea that fly CYRI can regulate actin dynamics through altering WAVE and Rac activity. They have resolved many of my earlier concerns, and in particular have better evidence now that the mutant phenotypes they show are due to loss of cyri function, and that CYRI functions cell-autonomously in border cells for their cohesion and collective movement and has a genetic interaction with WAVE. Thus, I do not think any additional data are required. Overall this is an interesting, well-executed and very detailed study that provides new evidence for Cyri function in cell collectives.

Some minor suggestions -

I think the statement on line 495 "Defects in border cell cohesion ultimately explain the partial sterility observed in cyri mutant females." is probably still too strong. It is a possible explanation, but the sterility is much more penetrant/severe (about 5 fold worse) than the border cell migration defects (affecting 40%), the number of border cells is not affected, and it has not been shown directly (or previously) that defects in the cohesion result in poorer fertility in general. So, they may want to revise this statement.

The presentation of Figure 3A is confusing since the gene at the top runs right to left but the sequences at the bottom run left to right- maybe the top one could be flipped to match.